# High resolution humidity profiles retrieved from wind profiler radar measurements

Frédérique Saïd[1], Bernard Campistron[1], and Paolo Di Girolamo[2]

[1]Laboratoire d'Aérologie, Université de Toulouse, UMR CNRS 5560, Toulouse, France
[1]Scuola di Ingegneria, Universita degli Studi della Basilicata, Potenza, Italy

*Correspondence to:* F. Saïd et al. (frederique.said@aero.obs-mip.fr)

**Abstract.**

The retrieval of humidity profiles from wind profiler radars has already been documented in the past 30 years and is known neither to be straightforward and nor as robust as the retrieval of the wind velocity. The main constraint to retrieve the humidity profile is the necessity to combine measurements from the wind profiler and additional measurements (such as observations from radiosoundings at a coarser time resolution). Furthermore the method relies on some assumptions and simplifications that restrict the scope of its application. The first objective of this paper is to identify the obstacles and limitations and solve them, or at least define the field of applicability. To improve the method, we propose to use the radar capacity to detect transition levels, such as the top level of the boundary layer, marked by a maximum in the radar reflectivity. This forces the humidity profile from the free troposphere and from the boundary layer to coincide at this level, after an optimization of the calibration coefficients, and reduces the error. The resulting mean bias affecting the specific humidity profile never exceeds $0.25$ g kg$^{-1}$. The second objective is to explore the capability of the algorithm to retrieve the humidity vertical profiles for an operational purpose by comparing the results with observations from a Raman lidar.

## 1  Introduction

Over the last 30 years, several authors (Gossard et al., 1982, 1998; Tsuda et al., 2001; Bianco et al., 2005; Stankov et al., 1996, 2003; Furumoto et al., 2003, 2007; Klaus et al., 2006; Imura et al., 2007) discussed the possibility of determining the magnitude of the humidity gradient profiles from measurements of zeroth, first and second moments of wind profiler radars (WPR) Doppler spectra either in the Ultra High Frequency (UHF) or the Very High Frequency range (VHF). The method exploits the clear air scattering properties of electromagnetic waves, this depending on the refractive index, and consequently on the thermodynamic properties (pressure, temperature, humidity) of clear air. In addition, most of these authors demonstrated the possibility to retrieve humidity profiles, by combining radar measurements with simultaneous measurements from other sensors, i.e. in situ radiosonde observations at a poorer time resolution (Tsuda et al., 2001; Stankov et al., 2003; Furumoto et al., 2006) or remote measurements, such as temperature profiles from a radio acoustic sounding system (RASS) (Tsuda et al., 2001), observations from a microwave radiometer profiler (Stankov et al., 1996; Gossard et al., 1998; Bianco et al., 2005; Klaus et al., 2006), precipitable water vapor measurements from a global positioning system (GPS) receiver (Gossard et al., 1999;

Imura et al., 2007), or a combination of these measurements. The method was considered to be promising for an operational implementation, providing the benefit of a finer time resolution, compared to the conventional radiosonde observations.

However, as far as we know, no successful attempt to apply this method to operational observations has been reported in literature. The first hurdle is related to the missing self-consistency of the method and the required synergy between various instruments. Another difficulty is represented by the necessity to carry out accurate measurements of the first three radar Doppler spectra moments, with an accurate calibration of the radar-backscattered power (zeroth order moment) and a careful post-processing of the radar observations. The latter is needed to guarantee that the velocity and width of the spectral peak (first and second order moment, respectively) are not disturbed by external contamination such as ground clutter, radio frequency interferences, spurious echoes like birds, etc. In addition, the relationship between the refractive index and the radar reflectivity is valid only in clear air conditions and this may raise difficulties when rainfall is mixed with clear air. [1] By contrast, smaller and saturated particles are considered as cloud particles. Radars emitting around 1.3-GHz (UHF) are more sensitive to rainfall than those emitting around 440-MHz (UHF) or 45-MHz (VHF) and require a careful processing to separate multiple-peaks signals in the vicinity of rainfall events. The final hurdle is represented by the fact that the procedure to determine the humidity profile from the humidity vertical gradient measured by the WPR is not straightforward and the different steps of the process are typically a source of accumulated errors.

In this study, we use WPR observations at 1274-MHz (UHF) collected under a variety of atmospheric conditions at mid-latitudes locations, and discuss how we cope with the above mentioned difficulties. UHF measurements are combined with consecutive radiosonde observations, spaced 6 h or longer, to retrieve high-resolution atmospheric humidity profiles. Section 2 provides the theoretical background for the algorithm used to retrieve the humidity profile which is based on the method developed by Tsuda et al. (2001). In our modified approach, we introduce a new constraint to integrate the humidity gradient vertical profile, which is the level of transition identified by a reflectivity maximum. This level corresponds either to the top of the mixed or residual layer, or to a temperature and/or humidity discontinuity. This approach modification revealed to be crucial to improve the quality of the results. Section 3 illustrates the experimental sites, the encountered meteorological conditions, the instruments involved in the study and the data processing applied on the WPR measurements. Section 4 illustrates the results of our algorithm under three categories of atmospheric conditions (encountered in the three different datasets) and discusses the calibration coefficients used for each dataset, in relationship with the coefficient values found in the literature. Finally, in section 5, we apply our method to the retrieving of high-resolution humidity profiles between 2 consecutive radiosonde observations and discuss the possibility to use it as an operational product. For one specific dataset we took advantage of the measurements of a ground-based Raman lidar located close to the WPR, to validate the high-frequency (15 min) humidity profiles retrieved by our algorithm.

---

[1]We consider as rainfall the droplets whose size is large enough to be detected by WPRs (typically larger than 100 $\mu$m) (Ralph (1995)).

## 2 Theoretical background

The refractive index of the air $n$, or refractivity $N = (n-1)10^6$, depend on atmospheric thermodynamic properties (Gossard and Sengupta, 1988):

$$N = 77.6 \frac{P}{T} + 3.73 \ 10^5 \ \frac{e}{T^2} \tag{1}$$

where $T$ is temperature $(K)$, $P$ is atmospheric pressure (hPa) and $e$ is water vapor partial pressure (hPa). We can also express $N$ in terms of specific humidity, $q$ (kg of water vapor per kg of moist air). $q$ is the parameter we aim at retrieving in the present work. Using the approximation $q = 0.622 \frac{e}{P-0.378e} \simeq 0.622 \frac{e}{P}$, Eq. (1) becomes:

$$N = 77.6 \frac{P}{T} + 5.99 \ 10^5 \ \frac{Pq}{T^2} \tag{2}$$

The vertical gradient of refractivity $M$ $(\text{m}^{-1})$ can be calculated by applying the following linearized equation for small perturbations (Gossard and Sengupta, 1988):

$$M = \frac{\partial N}{\partial T} \frac{dT}{dz} + \frac{\partial N}{\partial q} \frac{dq}{dz} \tag{3}$$

where

$$\frac{\partial N}{\partial T} = -77.6 \frac{P}{T^2} - 1.2 \ 10^6 \ \frac{Pq}{T^3} \tag{4}$$

$$\frac{\partial N}{\partial q} = 5.99 \ 10^5 \ \frac{P}{T^2} \tag{5}$$

Equation (3) leads to:

$$M = \frac{dN}{dz} = -77.6 \frac{P}{T^2} \frac{dT}{dz} - 1,2 \ 10^6 \ \frac{Pq}{T^3} \frac{dT}{dz} + 5.99 \ 10^5 \ \frac{P}{T^2} \frac{dq}{dz} \tag{6}$$

which shows that the refractivity vertical gradient consists of 3 different terms: the temperature gradient term (term 1), the humidity term (term 2), the humidity gradient term (term 3). As underlined by Tsuda et al. (2001), term 1 may be dominant under dry conditions such as winter conditions at mid-latitudes or in the upper atmosphere. They also found that term 2 usually contributes less than 10 %, even close to the surface where $q$ is the larger. The dominant contribution of term 3, depending on $\frac{dq}{dz}$ allows to solve very easily the first-order differential Eq. (6) (Tsuda et al., 2001). Some authors (Gossard et al., 1998; Stankov et al., 2003) assume that the partial derivatives in Eq. (5) are constant and can be estimated from standard atmosphere profiles. This assumption also imposes to neglect the contribution by the second term. In this paper we do not make this assumption, especially because one of our datasets is characterized by relatively moist conditions near the surface, which

imposes to consider all 3 terms in Eq. (6). As also demonstrated by Tsuda et al. (2001), the differential Eq. (6) can be solved after introducing the Brunt-Vaïssala frequency $N_{BV}$ (s$^{-1}$) given by:

$$(N_{BV})^2 = \frac{g}{\theta}\frac{d\theta}{dz} = g\frac{dln\theta}{dz} \tag{7}$$

where $\theta = \left(\frac{1000}{P}\right)^{2/7}$ and $g$ are the air potential temperature ($K$) and the acceleration of gravity (9.8 m s$^{-2}$), respectively.

This leads to:

$$M = 5.99 \ 10^5 \ \frac{P}{T^2}\frac{dq}{dz} - 1.2 \ 10^6 \ \frac{Pq}{T^2}\frac{(N_{BV})^2}{g} - 77.6 \ \frac{P}{T}\frac{(N_{BV})^2}{g} \tag{8}$$

which can be rewritten (Tsuda et al., 2001) in the form:

$$\frac{dq}{dz} + A(z)q = B(z) \tag{9}$$

where:

$A(z) = \quad -2\frac{(N_{BV})^2}{g},$ $\tag{10}$

$B(z) = \quad 1.65 \ \frac{T^2}{P}M + \frac{1}{7750}\left(\frac{dT}{dz}+\Gamma\right).$ $\tag{11}$

where $\Gamma = 9.8 \ 10^{-3}$ K m$^{-1}$ is the dry adiabatic temperature lapse rate. Tsuda et al. (2001) provided the following final form for Eq. (9):

$$q(z) = \theta^2 \int_{zo}^{z}\left(1.67 \ 10^{-6}\frac{MT^2}{P} + \frac{1}{7750}\frac{d\theta}{dz}\right)\theta^{-2} \ dz + q_o \tag{12}$$

with $q_o$ being the humidity at level $z_o$ where the integration is initialized.

Following Tsuda et al. (2001) also, Furumoto et al. (2006), Klaus et al. (2006) and Imura et al. (2007) used Eq. (12) to compute humidity profiles. We are also using this equation in the present work.

The next step is to relate $M$ to the radar characteristics. In clear air or cloudy conditions (precipitation-free atmosphere), UHF-range and VHF-range profilers detect the fluctuations of refractive index with a scale of one-half the radar wavelength,

through the following expression Ottersten (1969):

$$\eta = 0.38 \ C_n^2 \lambda^{-1/3} \tag{13}$$

where $\eta$ is the volume reflectivity for the turbulence echo (m$^{-1}$), depending on the radar return signal power, $\lambda$ is the wavelength and $C_n^2$ is the turbulence structure parameter (m$^{-2/3}$) for the radar refractive index. Gossard et al. (1982, 1998) found that, for homogeneous isotropic turbulence in a horizontally homogeneous medium with vertical gradients of mean properties, the squared vertical gradient of potential refractivity (potential refractivity is the value of $N$ for an air parcel moving adiabatically from its ambient level to the reference level -1000 hPa- without loss or gain of moisture) is:

$$\left(\frac{d\Phi}{dz}\right)^2 = \left(\frac{L_w}{L_\phi}\right)^{\frac{4}{3}} \left(\frac{dV}{dz}\right)^2 \frac{C_n^2}{C_w^2} \tag{14}$$

According to Tatarskii (1971) , $\frac{L_w}{L_\phi}$ is the ratio of two outer lengthscales, for shear ($L_w$) and for potential refractive index ($L_\phi$). Gossard et al. (1982) provided an empirical formulation for this ratio which they refined in Gossard et al. (1998). They found this ratio to be very small in stable layers and large in zones with near-neutral stability, with values ranging between 2 and 6. In this expression $\frac{dV}{dz}$ is the vertical shear of the horizontal wind vector :

$$\left(\frac{dV}{dz}\right)^2 = \left(\frac{du}{dz}\right)^2 + \left(\frac{dv}{dz}\right)^2 \tag{15}$$

where, $u$ and $v$ are the horizontal components of the wind. In Eq. 14, $C_w^2$ is the structure parameter of vertical velocity which can be expressed in term of the dissipation rate of the turbulent kinetic energy $\epsilon$ (m$^2$ s$^{-3}$) through the expression: $C_w^2 = \frac{4}{3} 2.1 \, \epsilon^{\frac{2}{3}}$ (where 2.1 is the Kolmogorov constant).

Assuming that $M^2 = \left(\frac{d\Phi}{dz}\right)^2$, Eq. (14) gets the form:

$$C_n^2 = \alpha^2 \, \frac{\epsilon^{2/3} M^2}{\left(\frac{dV}{dz}\right)^2} \tag{16}$$

Here, the first three moments of the radar spectral data are involved. Specifically, $C_n^2$ is related to the zeroth moment since the volume reflectivity $\eta$ in Eq. (13) is proportional to the backscattering signal power. The coefficient of proportionality $K$ between $\eta$ and the backscattering signal power depends on the radar properties including antenna efficiency, receiver bandwith, system noise power, losses in the transmission lines, etc. In most studies, $K$ is not known since the radar is not calibrated, so $K$ is included in the term $\alpha^2$ of Eq. (16).

The horizontal wind $V$ is determined from the first order moment, i.e. the Doppler shift in the spectral data, obtained at the end of the radar computation process. $\epsilon$ is related to the estimation of the second order moment, i.e. the broadening of the radar spectra.

Let us now discuss the coefficient $\alpha^2$. Assuming the radar to be calibrated, so that $K$ is known and the structure coefficient $C_n^2$ is an absolute value, Gossard et al. (1998) took a constant value for $\alpha^2$ of 0.44 regardless of the $M^2$ profile and the considered level in the profile. This value is obtained by comparing refractive index profiles obtained from balloon measurements to those estimated by a 440-MHz UHF radar, after an accurate post-processing of the first and second order moments of the spectral

data and an accurate radar calibration (that provided $K$). In fact, they measured an average value for $\frac{L_w}{L_\phi}$ of 4, which, after Gossard et al. (1982), should be considered as a conservative value under steady conditions.

Following Ottersten (1969), Gossard et al. (1982) and Stankov et al. (2003) used another expression for the turbulence structure parameter, based on the energy equation:

$$C_n^2 = a^2 \; \frac{\epsilon^{2/3} \; M^2}{\left(\frac{dV}{dz}\right)^2} \, (1 - R_f)^{-1} \qquad (17)$$

where $R_f$ is the flux Richardson number. They indicate that this relation is valid under fairly general assumptions inside regions of large kinetic energy transfer from shear into turbulence. A value of 2.8 is used for $a^2$.

For the specific case of a free troposphere which is most of the time hydrostatically stable and where the turbulence is known to be intermittent (in time and space), VanZandt et al. (1978) refined this relation by inserting $F$, a 'filling factor',

which accounts for the turbulent fraction of the backscattering volume. Based on the introduction of this factor, the expression for the turbulence structure parameter gets the form:

$$C_n^2 = 2.8 \; \frac{\epsilon^{2/3} \; M^2}{\left(\frac{dV}{dz}\right)^2} \, (1 - R_f)^{-1} \; F^{1/3} \qquad (18)$$

VanZandt et al. (1978) provided an estimation of $F$ based upon a simple model for the statistical distribution of wind shear and potential temperature gradient. The comparison of Eq. (16) and Eq. (18) reveals that $\alpha^2$ should depend on the stability

conditions in both the atmospheric boundary layer (ABL) and free troposphere, and also on the 'filling factor' in the free troposphere. In agreement with the concept of 'filling factor', Gossard et al. (1999) noticed that $\alpha^2$ could depend on anisotropy or turbulence unsteadiness. Tsuda et al. (2001) considered that $\alpha^2$ is not constant and requires calibration, even if the radar calibration coefficient $K$ is known. In the present work, we will pay special attention to the coefficient $\alpha^2$.

To summarize, the first three moments of the radar spectrum allow to determine $M^2$ (Eq. (16)), provided that $\alpha^2$ is known.

The sign of $M$ remains however ambiguous. In most cases it is negative, but it can become positive under clouds or in locally dry layers. Supposing that the ambiguity is resolved, the computation of $q$ through Eq. (12) requires the knowledge of the initial condition $q_o$ and of the potential temperature and pressure profiles (or at least an estimation of the air density at the surface to derive the pressure from the temperature profile). As underlined by Klaus et al. (2006), the estimation of a single humidity profile requires simultaneous measurements of 4 data-inputs, beside the radar measurements.

As already mentioned in the introduction, several authors proposed to get the temperature profiles from simultaneous RASS observations, and the initial density and humidity conditions by a meteorological station at the surface. This however raises an issue since the first radar gate is not at the surface and since the signal-to-noise is sometimes disturbed at the lower gates.

Other authors used the combination of radar wind profilers and ground-based microwave radiometers, the latter providing both temperature and humidity profiles, but at a coarser vertical resolution than the profiler vertical resolution. The first ad-

vantage of this method is represented by the possibility to calibrate the radar estimate of the $\mid M \mid$ value using the radiometer integrated humidity on the air column. A second advantage is represented by the possibility to determine the sign of $M$ from

the radiometer humidity profile, as in fact this sign is the same as the sign of the humidity gradient, or negative if the humidity gradient is negligible.

The integrated value of humidity on the air column can also be obtained from the GPS data and is sometimes used as a constraint for the calibration of $| M |$, provided that at least one in-situ humidity profile is available (e.g. from a radiosonde) to calculate the amount of the integrated humidity in the height range where the profiler identifies echoes relative to the total column integrated humidity. This method also assumes that this contribution is constant, which is a rough hypothesis, difficult to fulfill, especially because of the variability in the radar detection height.

Finally, another method consists in using low time resolution radiosonde profiles (6-hour or 9-hour spaced) to initialize $q$, calibrate $M$ and provide the temperature and pressure profiles. Then the objective with the radar is to provide intermediate profiles with a finer time resolution. This is the method we decided to use in the present work. In addition, we propose two refinements. Usually, the integration shown in Eq. (12) is initialized at the lowest gate measured by the radar, or, if the latter is too high and observation not available, at the upper boundary where $q$ becomes nearly zero (Tsuda et al., 2001). Here, we make two integrations. The first is started near the first radar gate, which is between 150 and 375 m above ground level (AGL) according to the radar and the measurement mode (see next section), while the second is started at the upper boundary of the radar profile, which is never the same since it depends on the backscattered echo power (the detectability is enhanced under moist conditions and reduced under very dry conditions). Both integrated profiles are adjusted (by joining the points) at a characteristic height that we will call Hlim. Hlim may correspond to the top level of the ABL, under unstable conditions, or at least to a strong moisture gradient under neutral or stable conditions. This level is easily detected by the WPRs, since it corresponds to a local maximum in the radar reflectivity profile. Hlim has been used for the past 20 years to monitor the depth of the mixed ABL (Angevine et al., 1994; Heo et al., 2003).

Since this level can demarcate two regions of the low troposphere, potentially characterized by drastically different turbulent conditions, we decided to compute $\alpha^2$ independently in both part of the profiles: below and above Hlim. $\alpha^2$ was determined in each of these regions by computing $M_{RS}^2 / M_r^2$. $M_{RS}^2$ is calculated from the radiosonde data through Eq. (6) and $M_r^2$ from the WPR data through Eq. (16). This calibration step could have exempted us from calibrating the WPRs. Finally, it is to be pointed out that, in our method, we also use the sign of the balloon humidity gradient to determine the sign of $M$.

## 3 Experiments, sites, instruments and radar data processing

### 3.1 Field experiments, measurement sites, instruments and radar data processing

Data were collected during three field experiments characterized by drastically different atmospheric conditions. The first experiment, the Boundary-Layer Late Afternoon and Sunset Turbulence (BLLAST) (Lothon et al., 2014), took place during the summer period in June 2011, at the Lannemezan Atmospheric Research Center (43.13°N, 0.13°E, elevation 595 m) at the Pyrenean foothills, and was characterized by typical fair weather convective boundary layers conditions found at mid-latitudes. The second campaign is the first Special Observing Period of the Hydrological cycle in Mediterranean Experiment (HyMeX SOP1), which took place over the western part of the Mediterranean basin during autumn 2012 (Ducrocq et al., 2014). In

this case the radar was installed in an atmospheric 'supersite' located in Candillargues (43.6°N, 4.07°E, elevation 2m) in the proximity of the seaside. The aim of the experiment was to study the upstream dynamical conditions linked to the initiation of strong rainfall events inland. The preferred situations were those when warm, moist and unstable air masses were advected from the sea. The third experiment was the second Special Observing Period of HyMeX (HyMeX SOP2) which was held during winter 2013, to study the mechanisms of air-sea exchanges in case of strong offshore winds (Estournel et al., 2016). The radar site was the same as for HyMeX SOP1.

## 3.2 Instruments

During these three experiments, intensive radiosounding (RS) operations (3h- or 6h-spaced) were performed with the purpose to monitor the atmospheric diurnal cycle. MODEM M2K2 and Vaisala RS92 (OYJ DigiCORA V3.64 / RS92-SGP) radiosondes were used for BLLAST and HyMeX, respectively. The accuracy of the RS92 humidity measurements has been assessed by Miloshevich et al. (2006), based on the comparison of these measurements with those simultaneously performed by a reference sensor of known abolute accuracy (a cryogenic frostpoint hygrometer) deployed on the same balloon. In the low troposphere, as considered in the present study, they found that the mean accuracy in the relative humidity measurements (with respect to the absolute sensor measurement) is always lower than 5 %. In 2010, the World Meteorological Organization conducted a new intercomparison experiment in China (Nash et al., 2011). According to these authors, Vaisala RS92 version tested in China showed a systematic error of less than 2 % and a random error of $\simeq$ 5 % in relative humidity measurements in daytime and nighttime, from the surface to the low stratosphere, in clear-air or cloudy conditions. During this experiment, the MODEM M2K2 radiosonde was also compared to a set of different radiosondes, including the Vaisala RS92. According to Nash et al. (2011), MODEM nighttime measurements had large positive biases (larger than 10 %) for most of the time in the lower and middle troposphere. MODEM radiosondes were also found to suffer from evaporative cooling when emerging from a cloud layer, and to overestimate relative humidity by 6 % at night, for relative humidities in the range 90-100 %, i.e. in cloudy air. Luckily the BLLAST measurements, in contrast with the HyMeX observations, were systematically carried out in clear air, which mitigated the uncertainties in the MODEM measurements.

WPRs in the framework of BLLAST and HyMeX were deployed by the Centre de Recherches Atmosphériques (Laboratoire d'Aérologie) and the Centre National de Recherches Météorologiques (Météo-France), respectively. Both profilers are 5-beam model PCL 1300 manufactered by Degreane. A detailed description of the WPRs, their main working parameters and the processing methods are provided in Saïd et al. (2016). Both radars were operated almost continuously at both sites over long periods (exceeding 18 months). For the purpose of this work, we will concentrate on the data collected from 19 June to 6 July 2011 during BLLAST, from 13 September to 5 November 2012 during HyMeX SOP1 and from 1 February to 15 March 2013 during HyMeX SOP2. Two operational modes were considered. The first mode (low mode), associated with a pulse length of 0.5/1 $\mu$s, sampled the lower troposphere from 75/150 m AGL to 5/5.7 km AGL (for BLLAST/HyMeX, respectively). The second mode (high mode) was specified for higher altitude sampling, from 150 m to 8 km AGL, and a pulse length of 2.5 $\mu$s was considered for both experiments. The vertical resolution was 75 m/150 m (for BLLAST/HyMeX, respectively) in low mode and 375 m in high mode. The high mode was oversampled and provided data every 150 m. The interpulse period (IPP)

was 40/45 $\mu$s (for BLLAST/HyMeX, respectively) in low mode and 80 $\mu$s in high mode for both experiments. The radar beam was steered into five directions: one vertical and four oblique directions at zenith angle of 17° and 90°-spaced azimuths. The beam width was 8.5°, narrow enough to enable accurate measurements of the Doppler spectral width in the low troposphere.

In the frame of HyMeX SOP1, the ground-based University of BASILicata Raman lidar system BASIL (Di Girolamo et al., 2009) was deployed in Candillargues and operated from 5 September to 5 November 2012, collecting more than 600 h of measurements, distributed over 51 measurement days and 19 Intensive Observation Periods (Di Girolamo et al., 2016). BASIL makes use of a frequency tripled Nd:YAG laser source, emitting pulses at 355 nm, with a single pulse energy of 500 mJ and a pulse repetition frequency of 20 Hz (average power at 355 nm: 10 W). The receiver consists of a Newtonian telescope (primary mirror diameter: 45 cm, f/2.1). The major feature of BASIL is represented by its capability to perform high-resolution and accurate measurements of atmospheric temperature and water vapor mixing ratio (kg of water vapor per kg of dry air), both in daytime and nighttime, based on the application of the rotational and vibrational Raman lidar techniques, respectively (Di Girolamo, 2004; Di Girolamo et al., 2009). Besides temperature and humidity, BASIL can also provide measurements of particle backscatter, extinction and depolarization at several optical wavelengths. Based on an integration time of 5 min and a vertical resolution of 150 m (which are the resolutions of the lidar data used in this paper), the typical daytime precision in water vapor mixing ratio measurements is 0.2 g kg$^{-1}$ up to 3 km and 0.3 g kg$^{-1}$ up to 5 km, while the typical nighttime precision is 0.05 g kg$^{-1}$ up to 3 km and 0.005 g kg$^{-1}$ at 10 km (Di Girolamo et al., 2016). In the following, we will use the symbol $q$ for both specific humidity and water vapor mixing ratio, since the percentage deviation between the two is rarely exceeding 1%, even in case of large humidity concentrations, which is far less than the systematic and statistical uncertainties affecting the lidar mixing ratio measurements. During HyMeX SOP1, BASIL measurements of humidity were calibrated based on the comparison with simultaneous radiosondes (the RS mentioned above) launched from a facility located approximately 100 m away from the lidar. A mean calibration coefficient was estimated by comparing BASIL and radiosonde data at all times when BASIL was running (approximately 50 comparisons).

### 3.3 Radar data processing

The radar time series were coherently averaged to reduce the computing time while preserving signal detectability. The number of coherent integrations (NCI) was calculated for each cycle of 10 beams (5-beam steering in two modes), according to the windspeed measured during the former cycle, to optimize the Nyquist interval. 128-point Fourier transforms of the finite time series were applied to obtain radial velocity spectra. Finally, an incoherent integration of 30 consecutive spectra was made to improve the signal detectability. In case of rainfall, IPP was enlarged, NCI changed and backscattered power reduced, to avoid second trace echoes or saturation. This first step in raw data processing was made with the software provided by the manufacturer and implemented in the radar sites. One full cycle was achieved every 4 to 5 minutes (according to NCI) and represented 12 to 15 minutes observations.

The second step, called consensus, was processed in real-time and also post-processed at the Laboratoire d'Aérologie. The purpose of the consensus data processing is to determine the meteorological spectral peak among the four more powerful peaks of the Doppler spectra at each range gate. The method is fully described in Saïd et al. (2016). Special care was devoted to the

separation between the meteorological peak and ground clutter echoes or the separation of individual echoes from a multiple-echo peak. This was decisive to provide an accurate spectral width of the Doppler peak, used to compute the dissipation rate of turbulent kinetic energy, $\epsilon$. Furthermore, special tests were performed to separate clear air spectra from precipitation spectra, using information from the four moments of the vertical velocity. Finally, we flagged out manually in the data post-processing the spectra disturbed by birds, that were frequent at night during the fall and late winter seasons of HyMeX, which coincided with migration periods.

## 3.4 Processing of the dissipation rate of turbulent kinetic energy

The dissipation rate of turbulent kinetic energy $\epsilon$ was computed according to the method and coefficients proposed by Jacoby-Koaly et al. (2002). Following Doviak and Zrnic (1984), Hocking (1988), Gossard et al. (1998) and White et al. (1999), they determined $\epsilon$ through the estimation of the broadening of the Doppler spectrum peak. The broadening of the spectrum had also to be corrected for contributions due to shear, to the antenna beamwidth and to the filtering effects of the Doppler spectrum. We chose to derive $\epsilon$ from a combination of the estimations obtained from the vertical velocity spectrum with the median of the estimations obtained from the oblique velocity spectra. This method had been previously assessed by Jacoby-Koaly et al. (2002).

## 3.5 Radar calibration

Both radar were calibrated according to the method proposed by Campistron and Réchou (2012) and improved by Campistron et al. (2013). The calibration is based on the comparison between rain rate measured by the profiler and raingauge at the ground. The height of the radar data is taken as low as possible considering signal saturation, receiver linearity, and ground clutter. Usually the best level is found around 600 m AGL. Long lasting stratiform precipitation periods were chosen to avoid the presence of strong vertical air velocities. Also high relative humidity periods were selected to minimize rain modification during its fall.

Following Ulrich (1983), Campistron et al. (2013) assume that raindrop size distribution follows a gamma function with two parameters that have to be determined. The drop fall speed in still air can be related to the droplets' diameter taking into account the change of density with height (Atlas et al., 1973; Foote and Du Toit, 1969). The parameters of the gamma distribution can be obtained using the mean vertical velocity and the radar reflectivity factor (Chu and Su, 2008). Finally the rainrate is derived by integrating the droplets' distribution over the diameter interval supposed to extend from zero to infinity.

During each of the rainfall events selected for the calibration, the radar constant was modified until the best agreement was found between raingauge and radar measurements. An average of the results was retained as the final calibration constant. As said before, the WPRs used in the framework of BLLAST and HyMeX had provided in fact longer datasets than the datasets we refer to for the present work. That is why we had no difficulty in finding several stratiform conditions to achieve the calibration. The calibration was done for the BLLAST and HyMeX radars in low mode, and for the BLLAST radar also in high mode. For each radar and each operation mode, the variability of the calibration coefficients $K$ induced by the calibration method never exceeded 12 %, which is small relative to the variability of the coefficients $\alpha^2$ that will be discussed further on (cf. Sect. 4.3).

## 3.6 Data conditioning for the humidity gradient retrieval

During BLLAST, the balloon took around 4 minutes to cross the ABL, whose top level was typically situated at level 2000 m above sea level (ASL) (1400 m AGL). Within the ABL, the wind remained weak (around 4 m s$^{-1}$), which corresponded to the preferred conditions for the experiment (clear air, anticyclonic conditions). A nocturnal low level jet sometimes occurred at night, but it was seldom stronger than 6 m s$^{-1}$. During the first 4 minutes, the balloon drifted horizontally by 1 km from the release site, located nearby the radar site (150 m). The balloon reached the typical maximum height of the BLLAST radar soundings (4 km) in 17 minutes (from the release), which corresponds approximately to three full cycles of the profiler. Therefore, for the humidity gradient comparison we used 15 minutes of radar data, during which the balloon drifted horizontally between 5 and 15 km from the radar site, according to the wind conditions in the free troposphere. At a level of 2 km, the horizontal coverage of radar observations was as small as 625 m, while it was twice as large at a height of 4 km.

During HyMeX SOP1, as anticipated before, the lowest atmospheric layers were mainly characterized by marine conditions, with southerlies or easterlies reaching 10 to 20 m s$^{-1}$ in the lowest 1-2 km (AGL and ASL). At this time of the year (fall season), the conditions were not propitious for sea-breeze development and mixed boundary layers had few opportunities to develop. Above 1 or 2 km, the wind conditions changed to the typical mid-latitude westerlies. The windspeed was never exceeding 25 m s$^{-1}$ at 5 km, which was the maximum height reached by the radar echoes during SOP1 (moister conditions than during BLLAST, so better detectability of the echoes). The balloons typically took 5.5 and 14.5 minutes to reach the heights of 2 km and 5 km, respectively. 14.5 minutes correspond to almost three full cycles of the radar. Due to the shear in the wind direction, the maximum drift of the balloon during its ascent was 10 km from its release point.

The radiosonde data were averaged by slices of 75 or 150 m (according to the radar vertical resolution), centered on the radar gates levels. This revealed to be a better choice than interpolating the radar data, but could yield some slight discrepancies in case of sharp humidity gradients.

We carefully interpolated the radar measurements to fill some gaps in the data. Profiles with gaps larger than 750 m were excluded from further analysis. Second order radar estimations (vertical shear and $\epsilon$) were smoothed to avoid sharp local derivatives. Since 1274-MHz WPRs are sensitive to both turbulence and raindrops, individual profiles were also checked to remove those contaminated by precipitation echoes (essentially during HyMeX SOP1). However, we kept the profiles for which the precipitation echoes were confined within a limited number of levels and the measured vertical velocity did not exceed 0.5 m s$^{-1}$.

## 4 Radar humidity profiles versus radiosonde profiles

### 4.1 Some adjustments to improve the method

Before providing detailed humidity profiles between consecutive radiosoundings, we had to check how the humidity profiles retrieved by the radar were consistent with the initial radiosonde profiles. This took the longest time to perform since it required several adjustments.

The first adjustment was already mentioned in section 2. We solved the sign ambiguity on $M$ (since the radar provides $M^2$), by assigning to $M$ the sign provided by the RS observations (Eq. (6)). Figure 1 illustrates $M$ and $q$ profiles before and after the correction. We had initially assigned a negative sign to $M$, which had revealed to be relevant for most of HyMeX SOP1 conditions, where the source of humidity is close to the surface (maritime air masses) and decreases with the height. There

were however situations, either during BLLAST or HyMeX, when drier layers disrupted this negative gradient. We show an example from BLLAST in Fig. 1. In Fig. 1 (a), $M$ is computed from Eq. (6) for the RS observations (red line) and extracted from Eq. (16) for the radar (black line), assuming a constant negative sign on the whole profile. The resulting radar humidity profile shown in Fig. 1 (c) rapidly deviates from the observed profile (RS profile) and maintains the deviation, in both the upper or lower parts of the profile. The downward and upward integrations computed to retrieve $q$ are connected at Hlim = 1745

m, which corresponds to a maximum in the radar reflectivity profile and to a change in the observed humidity profile. The approach to select this level will be discussed later on. The successive changes in the sign of $M$ are taken into account in Fig. 1b. The resulting radar humidity profile is clearly improved (Fig. 1 (d)). The thin red lines in Fig. 1 (c) and (d) represent the humidity retrieval as obtained from the integration of $M_{RS}$, after averaging the RS observations by slices of 75 m, to match the vertical resolution of the radar. The discrepancy between the two red lines can be considered as a systematic error associated

with the loss in vertical resolution linked to the 75 m averaging.

Occasional negative values of $q$ are put to zero, especially during BLLAST when very dry layers were observed. Similarly, some unexpected large values of $q$ were put to the humidity saturated value provided by the RS. This occurred during HyMeX SOP1 when moist conditions were frequently encountered. This limitation enabled to minimize the error accumulation in case of divergence of the integration. We will provide illustrations of such situations in the following.

Another improvement consisted in testing different values of the Hlim level in case of the presence of relative maxima in radar reflectivity profiles. An example is provided in Fig. 2. The $Cn^2$ profile in Fig. 2 (a) shows three peaks. The dominant peak at 902 m corresponds to the lower level of a cloud, since $q$ observed with the RS is saturated (the red solid and dashed lines are superimposed in Fig. 2 (b)). The double integration of the radar data upward and downward to this level leads to the radar profile of $q$ illustrated in Fig. 2 (b) (black line). Radar and RS profiles agree up to 1500 m, but above this level the radar

and RS profiles deviate. In this specific case the downward integration fails (below 3100 m). We recall that in the upper part of the profile, the integration is performed from 4800 m down to Hlim = 902 m. The combination of $Cn^2$, $\epsilon$ and shear measured by the radar fails in reproducing the discontinuity present in the RS profile between 3100 m and 2800 m. This may be due to the spatial heterogeneity of the air mass, but there is no specific ancillary information to prove it. Different values of Hlim (1802 and 3452 m) were also tested, based on the consideration of different peaks of the Cn2 profile. The best result is obtained

when Hlim = 1802 m, which is the height of the second peak from ground. The result is illustrated in Fig. 2 (c). It shows a slight improvement of the agreement in comparison to results shown in Fig. 2 (b), although the divergence of the radar profile below 3100 m down to 2400 m is still obvious. The transition at 1802 m does not correspond to any marked transition in the RS humidity profile. This choice probably improved the result due to its central location in the profile.

Figures 2 (d) to (f) illustrate the results for the same profile, when the low mode is used instead of the high mode. In this

case, the vertical resolution of the radar measurements is 150 m instead of 375 m. This can be checked on the $Cn^2$ profile in

Fig. 2 (d), which is not as smoothed as in Fig. 2 (a). The agreement between the radar and RS profiles of $q$ is sligthly improved, but again not striking (Fig. 2 (f) compared with Fig. 2 (c)).

Finally, another issue was raised by the choice of the initial conditions. We constrained the value of $q_o$ to be equal to the RS observation at the same level. This level sometimes coincided with a level of sharp humidity decrease, as illustrated in

Fig. 3 (a) at 3000 m or in Fig. 3 (c) at 4400 m. This was frequently the case at the upper boundary during BLLAST since the range gate where the radar echoes vanished also coincided with a sharp decrease of moisture. So we rectified the initial value by hand, to avoid an accumulation of the error along the whole profile, as shown in Fig. 3, where the humidity profiles prior (left column) and posterior (right column) to the correction are compared. We are well aware that this hand-made correction could not be applied for an operational purpose, but attempts to ingest this correction procedure in a dedicated algorithm are

currently underway.

## 4.2   Comparison between the RS and radar retrieved humidity profiles

We gathered in Figure 4 the results of the comparison between the radar estimation of the humidity profile and the observed RS humidity profiles for the three data sets in low mode. Panels (a), (c), (e) and (g) show the scatterplots of radar vs. radiosonde data, and panels (b), (d), (f) and (h) show the vertical profile of the deviations between radiosonde and radar data, all of them

obtained from low mode data. The red line in the scatterplots represents the linear regression line drawn from the data, which can be compared to the slope 1:1, i.e. the black line. HyMeX SOP1 data, the most numerous in terms of number of profiles, were split into two graphs. The vertical resolution of the BLLAST WPR measurements is three times higher than the one characterizing HyMeX measurements, which explains the larger sample size in Fig. 4 (a). Results are also summarised in Table 1, where the high mode has been added.

The best results are clearly those obtained during HyMeX SOP2, since the correlation coefficient is exceeding 0.93, the mean bias is 0.04 g kg$^{-1}$ or smaller and the mean standard deviation is 0.18 g kg$^{-1}$ or smaller. These successful retrievals are linked to the peculiar conditions of the HyMeX SOP2 experiment (limited variability of the humidity content). So we cannot rely upon this example to assess the effectiveness of our method. In most other cases, with the exception of the BLLAST data in high mode, the correlation coefficient exceeds 0.8, the slope of the regression lines is close to unity, the mean bias is equal

or smaller than 0.24 g kg$^{-1}$ and the standard deviation is not exceeding 1 g kg$^{-1}$ (Table 1).

We identified the reason why the BLLAST dataset gave poorer results in high mode than in low mode. In high mode these are 0.73, 0.36 g kg$^{-1}$ and 1.19 g kg$^{-1}$ for the correlation coefficient, mean bias and standard deviation, respectively (Table 1). These relatively lower quality results are due to the presence of thin layers of dry air (-300 m vertical depth) that the radar fails to reproduce in high mode since its vertical resolution (375 m) is not high enough.

The other results in Table 1 do not exhibit an outperformance at any of the two modes. Concerning the variability of the error and its standard deviation along the vertical, we cannot draw any general conclusion even if the HyMeX results seem to be better (smaller bias, smaller standard deviation) above 2500 m (Fig. 4, panels (d) and (f)). The bias remains small and similar for BLLAST within the whole profile.

To conclude, we consider that the method we propose yields good radar profiles to start the processing at a finer time resolution. Before getting the intermediate profiles, let us examine the variety of calibration coefficients $\alpha^2$ that were obtained for the three data sets.

### 4.3 Variability of the calibration coefficients

With the aim of comparing our estimates of the calibration coefficient $\alpha^2$ in Eq. 16 with values found in literature and studying its variability relative to the stability conditions, we first considered the coefficients within each dataset, which enabled to avoid possible issues about the radar calibration. According to Eq. 14 and 16 and considering the same WPR, the variability in $\alpha^2$ should reflect the variability of the ratio of the two outer scales $\frac{L_w}{L_\phi}$. Gossard et al. (1998) expected this quantity to be dependent on the stability conditions, at least inside the ABL, where the turbulence is homogeneous. Small ratios should occur

under stable conditions, which would correspond to large $\alpha^2$ values, and the opposite behavior under unstable conditions. To check this hypothesis, we used the radiosonde data to get an estimation of the stability conditions based on the gradient Richardson number :

$$Ri = \frac{g}{\theta_v} \frac{\frac{d\theta_v}{dz}}{\left(\frac{dV}{dz}\right)^2} \tag{19}$$

where $\theta_v$ is the virtual potential temperature. We chose the BLLAST dataset, since summer conditions provide the largest

range of stability conditions. Figure 5 (panel (a)) shows the calibration coefficients obtained from the BLLAST dataset in low mode as a function of the stratification with a differentiation of the lower part and upper part of the profiles, located below and above Hlim, respectively. The colorscale indicates the hour of the day. During the (dry) convective period of the day, the lowest portion of the profiles is expected to correspond to a mixed ABL, with Richardson numbers potentially negative in the surface layer, and close to zero above. The corresponding calibration coefficients can be clearly identified in the bottom left corner of

Fig. 5. Smaller values (below 0.01) are observed under unstable conditions, in accordance with Gossard et al. (1998). However a significant number of the coefficients corresponding to unstable conditions are also located in the range 0.03-0.5, where most of the coefficients of the lower layer can be found (the logarithmic mean of $\alpha^2$ in the lower layer is 0.11). On the contrary, nighttime coefficients measured within the lower layer (dark blue and orange circles) are preferentially large (>0.5), while the Richardson number can vary, but usually is close to or larger than 0.25, i.e. the critical value of the Richardson number,

indicating stable conditions. As expected, the upper layers are slightly or clearly stable (squares). In this case the logarithmic mean is 0.16, not far from the value observed in the lower layer. According to VanZandt et al. (1978), the variability of the coefficients in the upper part of the profiles is linked to the variation of the 'filling factor' and depends on the lapse rate of the free troposphere. Anyhow, this variability (the squares span roughly two orders of magnitude) is less marked than the variability due to the stability conditions of the low layers (the circles span four orders of magnitude). The average value of

the calibration coefficients for BLLAST is 0.13, which is close to the lower boundary of coefficients [0.26-1.11] proposed by Gossard et al. (1998).

The BLLAST coefficients obtained with the high mode are not reported since we estimated they are not representative, especially in the lower part of the profiles. First, the average of the transition levels (Hlim) is 1193 m AGL in high mode versus 928 m AGL in low mode, and the distribution is clearly shifted towards higher Hlim values in high mode, which means that the transition level is certainly higher than the boundary layer top in most cases. In second place, we obtained rather large values

of the coefficients (6 values larger than 10) for the lower part of the profiles, that we attribute to the poorer vertical resolution in high mode in an area of sharp variations of the humidity.

The same analysis was applied to the HyMeX datasets from which no clear result arose, neither during SOP1 nor SOP2 (not shown). During HyMeX, the development of the boundary layer was most of the time generated by mechanical turbulence (due to the wind intensity or to the roughness change at the sea/land transition). This can explain why the gradient Richardson

number is not a good indicator of the variability in the calibration coefficients under the HyMeX conditions. There is also no clear difference in the HyMeX calibration coefficient values between the lower and upper parts of the profiles, probably because moist convection equally affected all levels. Although the HyMeX radar was calibrated (in low mode), the coefficient values determined during SOP1 were three times smaller than those found during BLLAST. We artificially shifted the HyMeX calibration coefficients (by multiplying them by 3) to make the logarithmic average match, so to be able to compare the

variability. Results are illustrated in Fig. 5 (panel (b)) in terms of upper layer versus lower layer $\alpha^2$ coefficients. As seen before, due to the variability in stability conditions, BLLAST is characterized by a larger span of lower layer coefficient values. The calibration coefficient values determined during HyMeX SOP1 and SOP2 span roughly two orders of magnitude in the lower layers and two orders of magnitude in the upper layers on each side of the 1:1 line.

The main conclusion we can draw from these results, coming primarily from the BLLAST data, is the necessity of distin-

guishing between the mixed layer and the free troposphere in case of unstable conditions in the low troposphere. Fig. 5 (panel (b)) also suggests that the 'filling factor' proposed by VanZandt et al. (1978) for the upper layers do not vary that much. This explains why some authors obtained satisfying results by using a constant calibration coefficient value with VHF-band WPRs when sensing this portion of the atmosphere.

We propose an explanation for the different calibration coefficient values obtained during HyMeX SOP1 (0.04 on average)

and BLLAST (0.13 on average). We recall that both radars were calibrated. We assume that we can rely on this calibration since i) values of $Cn^2$ determined for BLLAST in high mode are similar to those obtained in low mode and ii) the two calibrations, based on the rain gauge measurements, are independent. During HyMeX SOP1, atmospheric thermodynamic conditions were most of the time close to saturation (as shown for example in Fig. 2, panel (c)). Although we discarded the rainy radar profiles when the rainfall affected the whole air column, it is likely that isolated pockets of rainfall may have locally

increased the radar $Cn^2$, which implies a decrease in $\alpha^2$ . We checked the distributions of $Cn^2$ for the 3 datasets and found that the logarithmic averages of $Cn^2$ (close to the median values) are 1.4, 31 and 1.0 $10^{-14}$ m$^{-2/3}$, for BLLAST, HyMeX SOP1 and HyMeX SOP2, respectively, which confirms the former hypothesis of higher $Cn^2$ values (and consequently lower calibration coefficient values) for the moist conditions during HyMeX SOP1. This disturbance can in principle be considered as a limit in the application of the method we propose, since the clear air turbulence conditions required for the application of

Eq. 13 is not totally fulfilled. However we consider that, based on the good results we obtained (see Fig. 4, panels (c) and (e)), the method can be successfully applied to HyMeX SOP1.

## 5 Continuous humidity monitoring between radiosonde observations

### 5.1 Method

Our objective in this section of the paper is to check whether radar data can be successfully used to describe the detailed structure of the low troposphere between two consecutive radiosoundings, even if the latter are 6 or 12 h-spaced. The two bordering soundings, called RS1 and RS2, are used as initial and final conditions. We first retrieved the radar humidity profiles closest to RS1 and RS2 on the bases of the application of the algorithm described in the previous sections. The algorithm provides the values of Hlim and the calibration coefficients at the launching times of RS1 and RS2. RS1 and RS2 also provided

the initial and final bottom and top border conditions for $q$, which we interpolated at the radar times (every 15 minutes). Pressure and temperature values were also interpolated at the times between RS1 and RS2 (at all levels) to provide $\theta$ and $P$ present in Eq. 12 and also to constrain $q$ to the humidity saturated value based on the interpolation of the RS data at the same time and level. In addition, the sign of $M$ at each level was taken as the sign of the RS1 humidity gradient at the same level for the first half of the period separating RS1 and RS2 and as that of RS2 for the second part of the period. Finally, the radar

calibration coefficients $\alpha^2$ were time-interpolated. In contrast, values of Hlim obtained from the radar data at the launching times of RS1 and RS2 were not interpolated. The radar provided an updated value of Hlim and an updated vertical profile of $M^2$ every 15 minutes (through new $C_n^2$, shear stress and $\epsilon$ profiles). Ultimately, using Eq. 12, the vertical profile of $q$ may be retrieved at a fine time resolution of 15 minutes. The choices for the parameters just described are summarized in Table 2.

    In case of several values of Hlim due to the presence of several peaks in the $C_n^2$ profile obtained at time $t$, a continuity

criterion was used to select the appropriate value. This criterion was applied to a variety of HYMEX case studies, which frequently showed multiple layers with sharp humidity gradients. By contrast, the $C_n^2$ profiles observed during BLLAST exhibited a marked isolated peak that could be directly taken as Hlim.

### 5.2 Some results

    A first example of the results obtained during HYMEX is shown in Figure 6. RS1 and RS2 are 6 hours-spaced and were chosen

to illustrate a case study, on 24 September 2012, when the moist lower troposphere dries and a mixed boundary layer develops. The RS1 humidity profile at 03:14 UTC is close to saturation from 500 m up to 2000 m (Fig. 6, panel (a)), whereas the RS2 profile at 09:00 UTC is drier and shows a mixed layer with a depth of 1350 m and a constant $q$ value of 8.4 g kg$^{-1}$ (Fig. 6, panel (b)). The humidity profiles retrieved by the radar between 03:14 and 09:00 UTC are represented with thin solid lines in Fig. 6 (panel (c)), with the time being color coded. The dashed lines are the corresponding saturated profiles calculated from the

profiles of $P$ and $T$ obtained by interpolating the data from the two RS. The radar profiles gradually dry in the lower levels and a mixed boundary layer develops from 07:14 to 09:00 UTC, accompanied by a decrease in air temperature (with a consequent

decrease of $q$ saturated between RS1 and RS2 in Fig. 6, panel (c)). In fact, the weak low-level wind, which had blown from the south in the early morning, turned gradually to north-westerly wind between the sunrise (around 05:00 UTC) and 06:14 UTC (not shown). The mist marine layer was then replaced by a continental, cloud free boundary layer that could easily develop due to an increase in the wind strength (15-20 m s$^{-1}$) and to a larger surface-air temperature contrast. The radar was particularly

helpful to detect the top of the mixed layer.

In Figure 6 (panel (d)), we compare the radar humidity profiles (solid lines) to the 15 min-averaged profiles obtained with the lidar at the same times (crosses) and to the humidity profiles calculated from a linear interpolation of the RS1 and RS2 profiles (dashed lines). The first four lidar observations (between 03:14 and 04:59 UTC) are attenuated above 750 m because of the presence of a cloud layer. In fact, atmospheric particles can lead to antagonistic effects on the lidar beam: few and scattered

particles may lead to an increase of the backscattered radiation, whereas dense particle ensembles, as those found in a thick cloud, are usually characterized by large optical thicknesses (>1-2), which translates into laser beam attenuation overwhelming particle backscattering. In the early morning, the sharp decrease of the lidar signal above 750 m in Fig. 6 (panel (d)) clearly reveals the base of a thick cloud at 750 m. As the air dries up with the time the lidar recovers its capacity to cover the lower troposphere and the final lidar humidity profiles are very close to the RS2 and radar profiles.

The representation in Fig. 6 (panel (d)), hardly enables to distinguish between the radar humidity and the humidity calculated from the RS interpolation. That is why we preferred to show the same results with height-time cross-sections of humidity (Fig. 7). In this figure, we highlight with black dots the levels characterized by saturation conditions. Due to the difficulty to rely on the lidar humidity profiles measured under saturated conditions, we superimposed on the lidar WVMR map in Fig. 7 (panel (a)), the dots obtained with the RS data (the same as those in Fig. 7, panel (c)). The thick cloud remains over the lidar until

05:00 UTC. Simultaneously, the radar detects saturated values of $q$ (Fig. 7, panel (b)). After a short period of clear air, another cloud is advected over the measurement site between 05:30 and 06:30 UTC, with a base at 1100 m, as indicated by the lidar. This cloud is also captured by the radar, with saturated values of $q$ from 1500 m to 2200 m, at 05:30 and 06:30 UTC. After 05:00 UTC the lidar shows humidity values similar to those measured by the radar and the RS in the low layers, although the mixing of the boundary layer is best represented by the radar and probably the most likely since it results from an accurate

measurement of the inversion height, which is well marked by the Hlim transition. The moisture retrieved by the radar exhibits a mixed boundary layer, whose depth increases from 500 m (at 07:00 UTC) to 1300 m (at 09:00 UTC) and dries with the time.

With the following example (case study on 23 September 2012 from HyMeX SOP1), we extended the time separating the two border radiosounding by choosing RS1 at 09:01 UTC and RS3 at 20:31 UTC. This enabled to use an intermediate RS at 14:58 UTC, called RS2, and to compute the radar profiles during three distinct periods, namely 09:01-14:58 UTC (panels (a),

(d), (g) and (j) in Fig. 8, 14:58-20:31 UTC (panels (b), (e), (h) and (k)) and 09:01-20:31 UTC (panels (c), (f), (i) and (l)). The juxtaposition of the first two columns should give the third. As expected, this is clearly the case for the lidar (panels (d), (e) and (f)) but not so obvious for the interpolated RS (panels (j), (k) and (l)).

The RS bottom border conditions in Fig. 8 (panel (l)) do not recreate the drying that occurs at 500 m between 09:00 and 15:00 UTC (panels (a) and (j), 12 down to 7 g kg$^{-1}$) and the following moistening from 15:00 to 20:30 UTC. The saturated areas

(black dots) in panels (l) or (j) and (k) are not consistent either. Remember that dots in this row here indicate the interpolated RS. In fact, the decrease in $q$ at 500 m in panel (j) widens the gap to the saturation, especially as the saturation values climb due to the increase of temperature with time. In contrast, the low layers moistening from 15:00 to 20:30 UTC rapidly leads to saturation conditions, not at the surface where the air is warm, but at a higher altitude of 1300 m (Fig. 8 (k) around 16:00 UTC). A deeper cloud layer appears around 18:00 UTC as testified by the RS interpolation in panel (k). The thickening of the cloud layer occurs later, around 19:30 UTC in panel (l). Consequently, a simple interpolation within the 12 h interval between RS1 and RS3 seems unrealistic since it does not reflect the proper daytime evolution of the low level boundary conditions. This also influences the saturation conditions at the mid-level.

The comparison with the lidar enhances the difficulty for the radiosoundings to describe the humidity fields under non steady atmospheric conditions. On 23 September 2012, a strong convective line was active, far west of the site, extending from Iceland to the west of the Iberian Peninsula (this convective line finally crossed the measurement site during the following night). This frontal activity generated a pocket of moist air in the lower layers, that was advected by south-easterly winds (150° at 500 m) over the western Mediterranean sea, while it circumvented the Pyrenean mountains. Another pocket of moist air was situated to the east of the site, due to some convective activity over Italy. Between these two moist areas, there was a pocket of drier air that moved according to the relative influence of the two convective areas on each side. Consequently, the measurement site, that was located in this area, encountered varying moisture conditions.

The lidar is well capturing the large humidity variability characterizing the lower levels with large amounts of $q$ (13-14 g kg$^{-1}$) in the time interval 09:30-13:30 UTC (Fig. 8 panel (d)), as a result of the influence of the western convective line, and the drying of the lower layers in the time interval 13:30-20:30 UTC (Fig. 8 panel (e)), when the moisture pocket moved further to the west. The 09:00 UTC RS that was launched during a period of increasing moisture (12 g kg$^{-1}$ between 500 and 800 m), is not able to capture the whole increase (unfortunately the lidar data were missing at 09:00 UTC as indicated by the blue vertical stripe from 09:00 to 09:45 UTC in Fig. 8 panel (d)). Consequently the radar, whose initial conditions are based on the 09:00 UTC RS, fails in capturing the large humidity amounts characterizing the lower layers between 09:30 and 13-30 UTC (Fig. 8 panel (g)).

Between 13:30 and 17:00 UTC, the radar estimations below 1500 m are closer to the lidar measurements. Both instruments indicate a top level of the moist layer varying between 1200 and 1500 m (Fig. 8 panels (d-e) and (g-h)). Above this level, the lidar beam is extinguished by the thick clouds. The detection of this transition level is facilitated, for the radar, by the fact of being marked by a large wind shear which favors the $Cn^2$ increase.

Between 17:00 and 18:30 UTC, the radar and the lidar both detect higher humidity amounts in an intermediate layer between 800 and 1500 m or 500 and 1100 m, respectively (Fig. 8 panels (e) and (h)). Within this layer, the radar shows saturated values close to the levels where the interpolated RS values are saturated, but limited between 17:00 UTC and 18:30 UTC. The radar humidity values are also consistent with the lidar values. Between 16:00 UTC and 18:30 UTC, the radar detects a cloud in the vertical region 2000-2500 m (Fig. 8, panels (b) and (h)). A cloud is also well visible in the particle backscatter field (not shown) obtained by the lidar. The particle backscatter data at 1064 nm are able to properly reveal both aerosol layers and

cloud/precipitation particles. Specifically, hydrometeors evaporating/sublimating before reaching the ground are observed (as vertical thin stripes) between 17:30 and 19:00 UTC in the vertical region 0.5-1.2 km.

Finally, if we now consider the capacity of the radar to retrieve the humidity variability during the whole period from 09:00 UTC to 20:30 UTC (Fig. 8, panel (i)), we must recognize that the radar fails to reproduce the lidar data (Fig. 8, panel (f)), with the exception of the transition level around 1400 m, between 13:00 UTC and 20:30 UTC. The cloud mentioned previously is unfortunately not seen, probably due to the erroneous calibration coefficients imposed by the too large interval separating the two bordering RS.

The 23 September 2012 case study showed that, for changing conditions near the surface (that are not typical of a classical diurnal cycle evolution) and 12h-spaced radiosoundings, both the combined radar-RS algorithm and the simple time interpolation between two radiosoundings fail to reproduce the detailed structure of the humidity in the low troposphere, essentially because of a lack of documentation of the evolution of the humidity at the first radar gate. As said before, the data from a ground station would not have been helpful, due to the variability between the surface and the first radar gate (see for instance in panel (e) of Fig. 8). By contrast, when the time interval between the two RS was reduced to 6 h, the radar proved to be able to retrieve the humidity amounts measured by the lidar and the presence of a cloud in the intermediate layers.

Another example to illustrate the performance of the radar-based approach with respect to a RS interpolation and the lidar measurements is shown in Fig. 9. RS1 and RS2 on 26 September 2012 are 6h-spaced. Both show clear air and a regular reduction of the humidity content with the height (Fig. 9, (panel (c)). In fact, a pocket of moister air occurred between the two RS at 03:00 and at 09:00 UTC, that is well observed with the lidar (Fig. 9, panel (a)), and partly retrieved by the radar (Fig. 9, panel (b)), with local saturated values. Even if the radar retrieval shows a few erroneous profiles (04:30, 05:30 , perhaps 07:00 and 07:30 UTC...), the radar documentation is necessary to detect the morning moistening of the air.

## 6 Conclusions

We demonstrated in the first four sections of this paper that, although WPRs, with their first three moments, measure essential parameters for the determination of the vertical humidity gradient, the radar data cannot be used to retrieve the vertical profiles of humidity independently from other sensors' data. To obtain the profiles, we applied a method already proposed by Tsuda et al. (2001), which consists in using a combined retrieval algorithm exploiting WPR measurements supported by RS observations at a coarser frequency. This algorithm is based on several approximations and assumptions that proved to be appropriate since the accuracy of the results we obtained did not exceed 0.25 g kg$^{-1}$ (mean bias between $q$ radar and $q$ RS). To obtain these results, we improved the algorithm proposed by Tsuda et al. (2001), by using a key parameter from the radar, which is the level of the reflectivity peak value, Hlim, allowing to split the calculations in two parts, with two different calibration coefficients accounting for two distinct vertical regions of different turbulence characteristics. The introduction of this level also mitigated the errors by replacing a long integration by two shorter ones.

After assessing the algorithm at the time of the RS observations, we applied it between two RS profiles, to obtain humidity profiles at a finer time resolution and to check the performance of the combined algorithm with respect to a simple RS time

interpolation. We used, when available, simultaneous lidar data to assess the results. The set of data that enabled this comparison was collected during a period seldom characterized by the presence of clear-sky conditions, while cloudy conditions were prevailing (HyMeX SOP1). In the presence of clouds, the lidar beam is rapidly attenuated above cloud base, so that the assessment can only be made in the lower portion of the profiles.

We obtained some satisfactory results, provided that the time separating the two boundary RS did not exceed 12 h. However we met also some hindrances that make the method hard to apply in an entirely automatic way, due to the assumptions we made. These difficulties are summarized below :

– The most restrictive issue is the one associated with the border conditions (bottom and top). The method assumes that they vary linearly between the two RSs which is not always true. If the border conditions are not well defined (for instance
at a level of strong moisture gradient), the error may propagate and become large at the Hlim level. Additionally, the resulting profiles can easily move apart, towards the two constraining borders: either towards $0$ g kg$^{-1}$ as the minimum value, or towards the saturated moisture content as the maximum value.

– Although pressure and temperature are secondary parameters in the algorithm, so that their estimate does not need to be as accurate as the border conditions, the profiles for these parameters have to be provided. These two parameters are also
used to constrain the computed humidity values to the saturation $q$. We used a linear interpolation of the two border RS to get the intermediate $P$ and $T$ profiles. Alternatively, these profiles could be provided by models, which are usually more reliable for pressure and temperature than they are for humidity.

– The constraint on the sign of the humidity gradient is also an issue that can hardly be solved by a simple interpolation or a continuity constraint in time. Some authors constrain their results with GPS measurements of the integrated water
content. This approach failed with our data set.

– We highlighted the necessity of calibrating the vertical gradient of radar refractivity, with calibration coefficients likely to vary in time and space. This revealed to be helpful, but also in this case, a simple interpolation between the initial and final coefficients could be too large an approximation. However, the detection of the transition level between the boundary layer and the free troposphere was definitely helpful.

Finally, we demonstrated that the combined RS-radar algorithm used to retrieve the humidity profiles outperforms a simple interpolation of the RS observations. The radar is especially skilled at determining the evolution of the transition layers, which is usually an issue when using other remote-sensing measurements such as, for example, radiometer measurements. However, the present method should be used with caution, and is probably more adequate in postprocessing a dataset for scientific purpose than for a blind use in an automatic platform.

*Acknowledgements.* Data were obtained from the HyMeX program, sponsored by grants from MISTRALS/HyMeX and ANR-11-BS56-0005 IODA-MED through the Fonds Européen de Développement Régional of the European Operational Program 2007-2013. BLLAST field experiment was made possible thanks to the contribution of several institutions and supports : INSU-CNRS (Institut National des Sciences de l'Univers, Centre national de la Recherche Scientifique, LEFE-IDAO program), Météo-France, Observatoire Midi-Pyrénées (University of Toulouse), EUFAR (EUropean Facility for Airborne Research) and COST ES0802 (European Cooperation in the field of Scientific and Technical). BLLAST field experiment was hosted by the instrumented site of Centre de Recherches Atmosphériques, Lannemezan, France (Observatoire Midi-Pyrénées, Laboratoire d'Aérologie). We also thank the CNRM team (Météo-France, Toulouse) for operating the Candillargues radar and launching the radiosoundings and SEDOO (Observatoire Midi-Pyrénées, Toulouse, France) for their help in quick-look providing and data-base management for both BLLAST and HyMex experiments.

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

**Table 1.** *Correlation coefficient $R^2$ of the linear regression between radar and RS humidity values for the three data sets. Mean bias (RS minus radar) and standard deviation for the whole dataset is also specified, together with the largest standard deviation value in the dataset. 'lm' and 'hm' stand for low mode and high mode, respectively.*

| dataset | mode | $R^2$ coef. | mean bias (g kg$^{-1}$) | mean std (g kg$^{-1}$) | max std (g kg$^{-1}$) |
|---------|------|------|-----------|----------|---------|
| BLLAST | lm | 0.87 | -0.07 | 0.82 | 1.49 |
| | hm | 0.73 | 0.36 | 1.19 | 1.58 |
| HyMeX | lm | 0.89 | 0.19 | 0.92 | 2.25 |
| September | hm | 0.92 | 0.02 | 0.70 | 1.69 |
| HyMeX | lm | 0.85 | 0.24 | 0.85 | 1.24 |
| October | hm | 0.87 | 0.08 | 0.98 | 1.20 |
| HyMeX | lm | 0.94 | 0.04 | 0.17 | 0.3 |
| February | hm | 0.93 | 0.01 | 0.18 | 0.32 |

White, A. B., Lataitis, R. J., and Lawrence, R. S.: Space and Time Filtering of Remotely Sensed Velocity Turbulence, Journal of Atmospheric and Oceanic Technology, 16, 1967–1972, doi:10.1175/1520-0426(1999)016<1967:SATFOR>2.0.CO;2, http://journals.ametsoc.org/doi/abs/10.1175/1520-0426%281999%29016%3C1967%3ASATFOR%3E2.0.CO%3B2, 1999.

**Table 2.** *Parameters used to retrieve 15 min-spaced vertical profiles of q in the time interval between two radiosoundings RS1 and RS2. Hlim serves to apply the convenient calibration coefficient and to merge the two integrals.*

| | |
|---|---|
| **Parameters used to calculate $M^2(z)$ in Eq. (16)** | |
| $C_n^2(t,z)$, $\epsilon(t,z)$, $\frac{dV(t,z)}{dz}$ : | provided by the radar every 15 min |
| $\alpha^2(t)$ lower layers : | from a 15 min-interpolation between the radar/RS1 and radar/RS2 calibration coefficients |
| $\alpha^2(t)$ upper layers : | from a 15 min-interpolation between the radar/RS1 and radar/RS2 calibration coefficients |
| **Parameters used to retrieve $q(z)$ from Eq. (12)** | |
| $\theta(t,z)$, $P(t,z)$ : | 15 min-interpolated profiles from RS1 and RS2 $\theta(z)$, and $P(z)$ |
| $q_o(t)$ bottom : | 15 min-interpolated $q$ from RS1 and RS2 at the lower common level |
| $q_o(t)$ top : | 15 min-interpolated $q$ from RS1 and RS2 at the upper common level |
| $Hlim(t)$ : | extracted from the radar $C_n^2$ profile every 15 min (usually the peak value) |
| constraint of sign of $M$ at level $z$ and time $t$: | depends on the sign of $M(z,t)$ (or humidity vertical gradient) for RS1 and RS2 |
| $q(t,z)$ saturated value (to constraint $q(t,z)$) : | from a 15 min-interpolation of $T(z)$ and $P(z)$ between RS1 and RS2 |

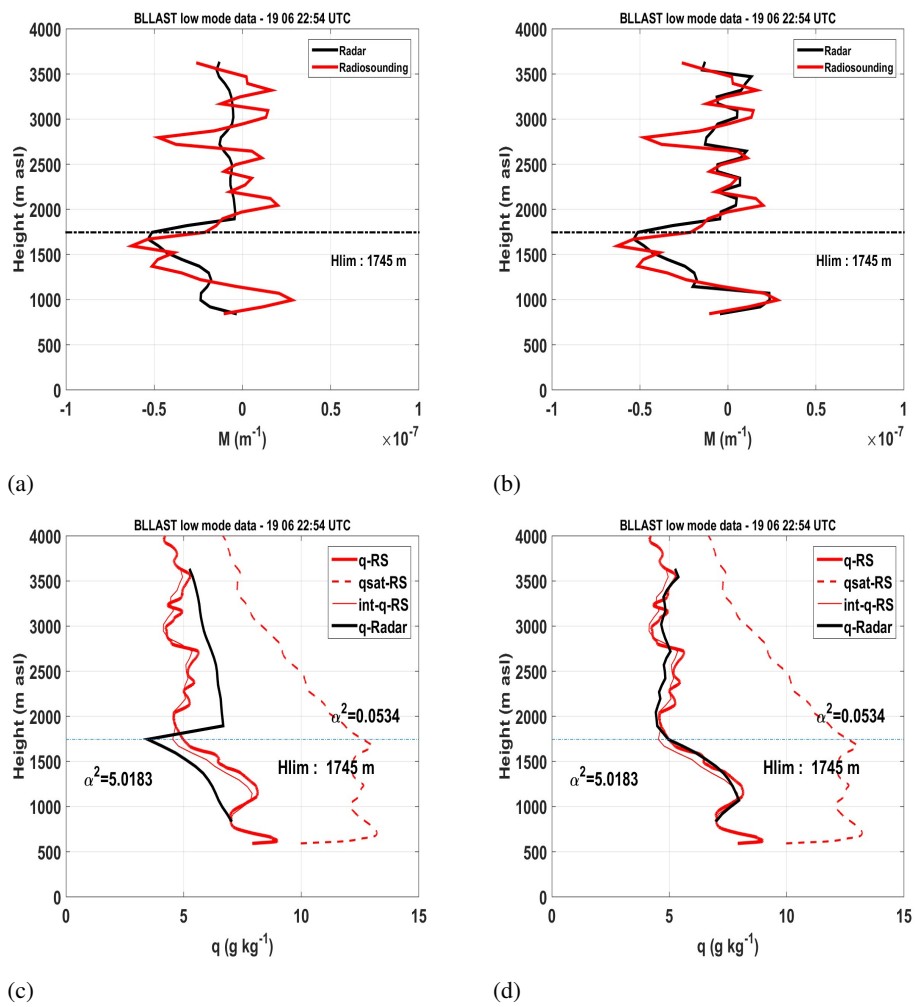

**Figure 1.** *Vertical profiles of refractivity gradient (panels (a) and (b)) and humidity (panels (c) and (d)) using a systematic negative sign for M (panels (a) and (c)), or after assigning to M the sign of M provided by the RS observations (panels (b) and (d)). RS values (q-RS) are red solid lines and radar values (q-Radar) are black solid lines. The thin red lines (int-q-RS) identify the humidity profiles retrieved from the integration of $M_{RS}$ after averaging the RS observations by slices of 75 m, to match the vertical resolution of the radar. The red dashed line (qsat-RS) is for the saturated humidity profile. The horizontal dashed line delineates Hlim, the transition level used to separate the upper and lower part of the profile. The two values of $\alpha^2$ (calibration coefficients), over or below Hlim are also indicated.*

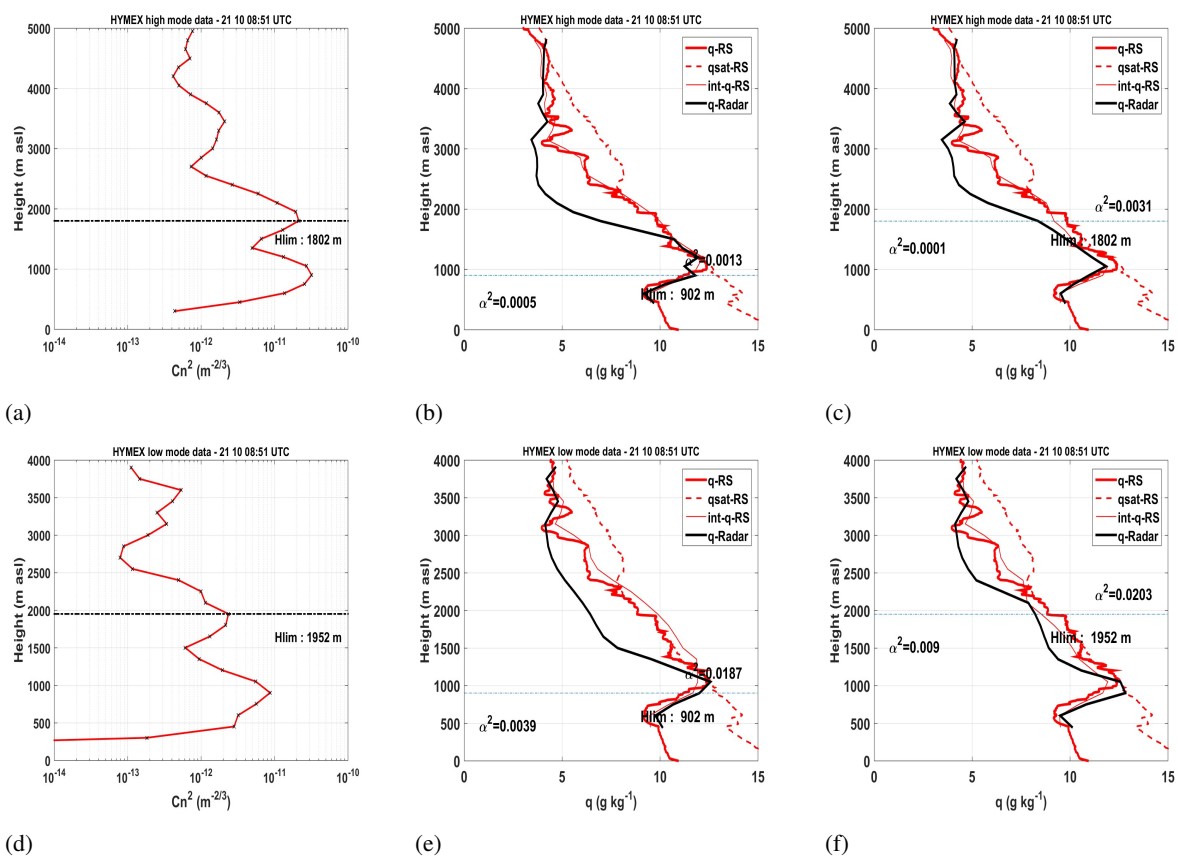

**Figure 2.** *Radar turbulence structure parameter $Cn^2$ (panels (a) and (d)) and humidity profiles (same details as in Fig. 1 (c)) for different Hlim levels. Panels (b) and (e) illustrate the WPR specific humidity profile (q-Radar) obtained considering the value of Hlim corresponding to the dominant $Cn^2$ peak observed in panels (a) and (d), respectively. Panels (c) and (f) illustrate the WPR specific humidity profile obtained considering the value of Hlim corresponding to a different relative maximum of the $Cn^2$ profile. In panels (b), (c), (e) and (f), the RS specific humidity profile (q-RS), the saturation specific humidity profile (qsat-RS) obtained from RS pressure and temperature profiles, and the saturation specific humidity profile (int-q-RS) obtained by integrating RS vertical gradient of refractivity (MRS) are shown. Panels (a), (b) and (c) are for the high mode (vertical resolution 375 m, interpolated every 150 m), while panels (d), (e) and (f) are for the low mode (vertical resolution 150 m).*

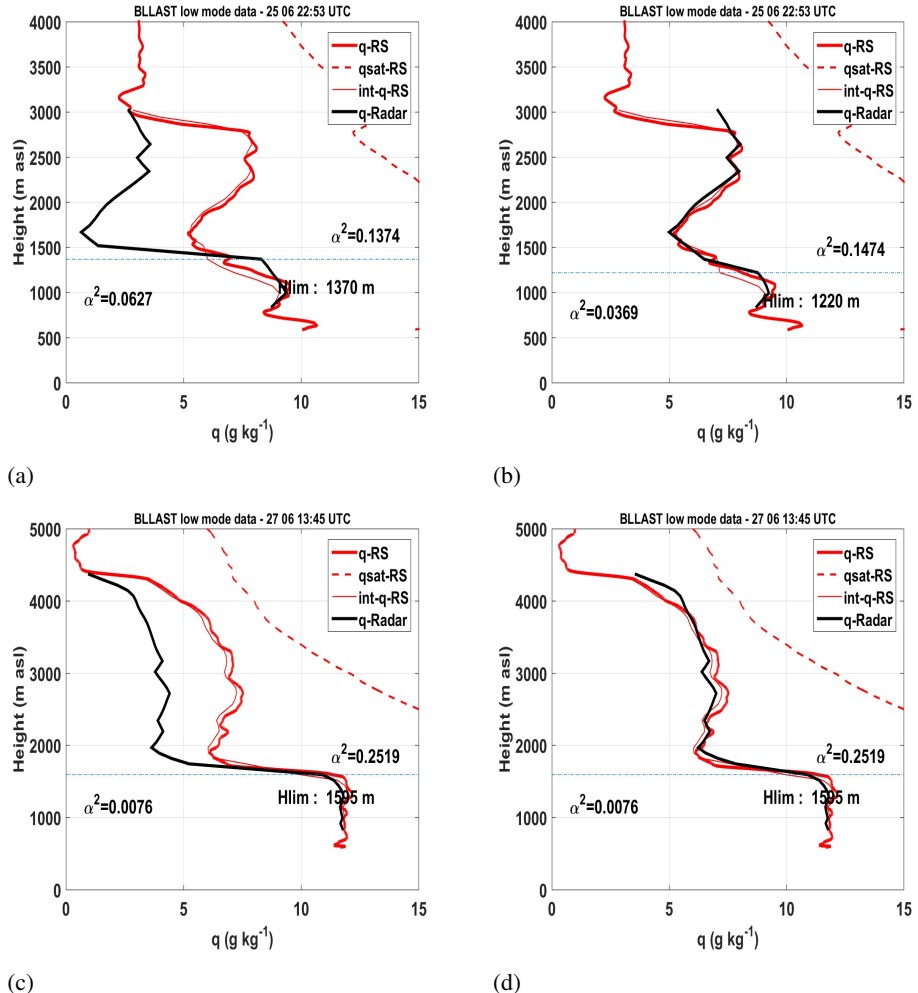

**Figure 3.** *Specific humidity profiles obtained: using an automatic detection of the initial specific humidity value ($q_o$) at the level $z_o$ where the integration is initialized (panels (a) and (c)); through an adjustment of $q_o$ (panels (b) and (d)), for two different profiles from the BLLAST data set. The vertical resolution is 75 m. Same details as in Fig. 1 (c).*

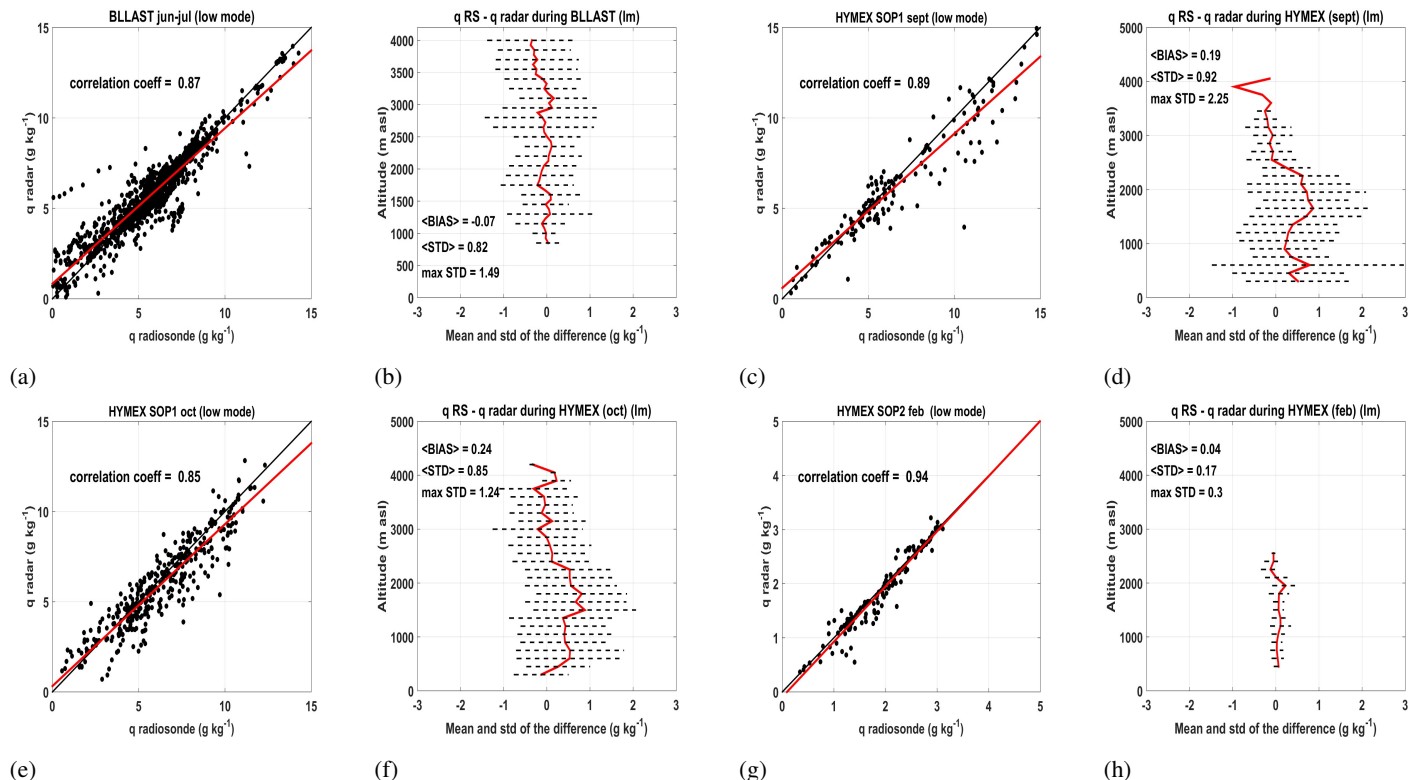

**Figure 4.** *Panels (a), (c), (e) and (g) : scatterplots of radar versus RS specific humidity during June and July 2011 (BLLAST), September 2012 (HyMeX SOP1), October 2012 (HyMeX SOP1) and February 2013 (HyMeX SOP2), respectively, with the linear regression line (in red) and the 1:1 slope line (in black). The $R^2$ correlation coefficient of the regression is also specified. Panels (b), (d), (f) and (h) : deviation profiles between the RS specific humidity measurements and radar-based estimate for the above specified data sets $\pm$ the standard deviation. The mean bias and standard deviation values for the whole dataset are reported, along with the maximum standard deviation value (g $kg^{-1}$).*

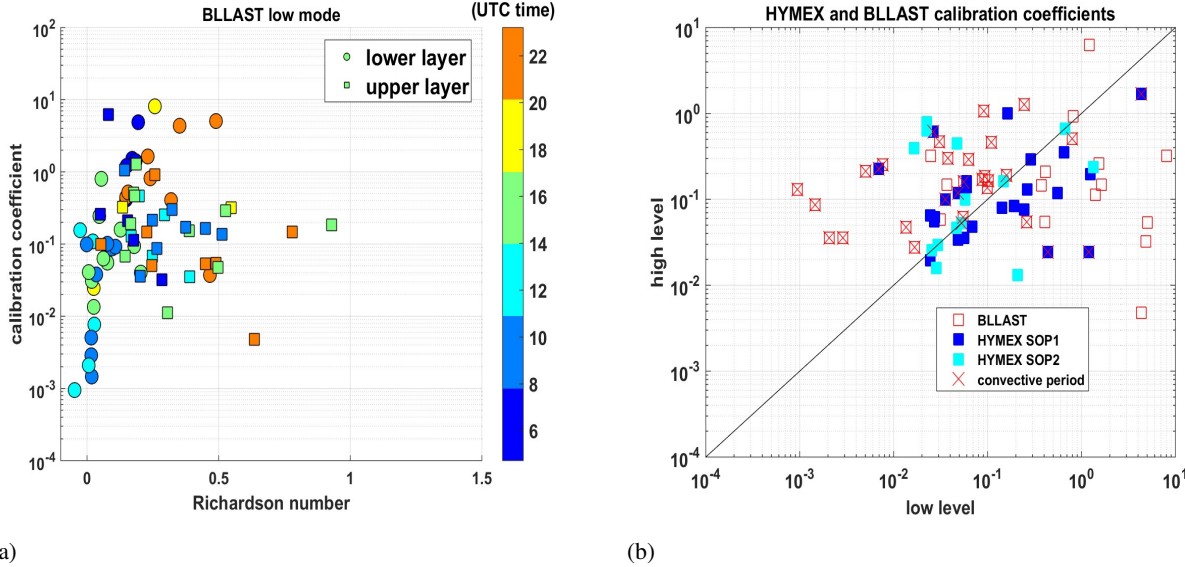

(a)                                                              (b)

**Figure 5.** *Panel (a): $\alpha^2$ calibration coefficients obtained with the BLLAST dataset in low mode as a function of the stratification estimated with the gradient Richardson number. The results are divided into the lower (circles) and upper part (squares) of the ABL and shown as a function of the time of the day (colored scale). Panel (b) : scatterplot of the calibration coefficients $\alpha^2$ (high level versus low level) obtained with the BLLAST and HyMeX datasets in low mode. The crosses correspond to the profiles measured during the potentially dry convective period i.e. 09:30-17:00 UTC.*

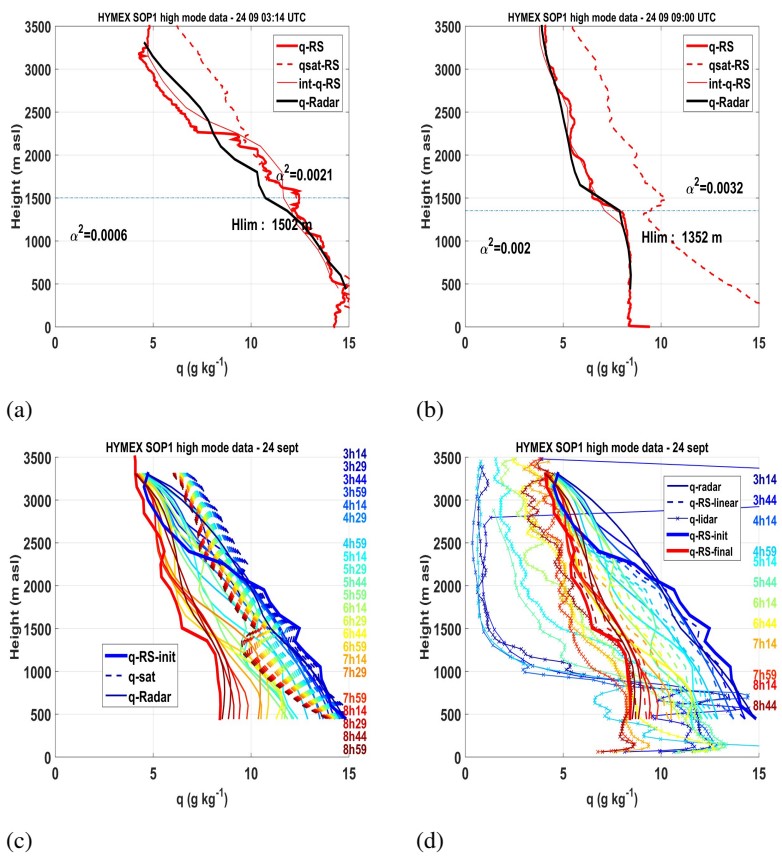

**Figure 6.** *Vertical profiles of humidity measured during HYMEX on 24 September 2012: the radar and RS profiles used as time border conditions are shown in panels (a) and (b) with the same details as in Fig. 1. These border conditions are reproduced in panels (c) and (d) with the thick blue solid lines for RS1 (03:14 UTC) and red one for RS2 (09:00 UTC). Intermediate radar profiles (thin solid lines) and corresponding saturated water vapor profiles (dashed lines) are presented in panel (c). Panel (d) shows the lidar humidity profiles (thin lines with crosses), the radar ones (thin lines) and the humidity profiles resulting from a linear interpolation between RS1 and RS2 (dashed lines).*

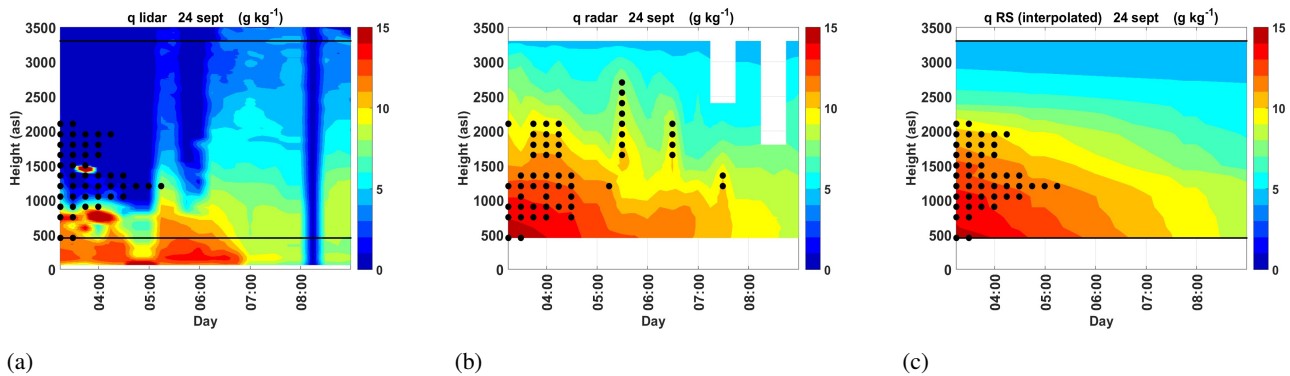

**Figure 7.** *Time-height cross sections of q observed on 24 September 2012 by the lidar (panel (a)), calculated from the radar data (panel (b)) and interpolated between RS1 and RS2 (panel (c)). The dots are for the radar saturated values in panel (b), and RS saturated values in panels (a) and (c).*

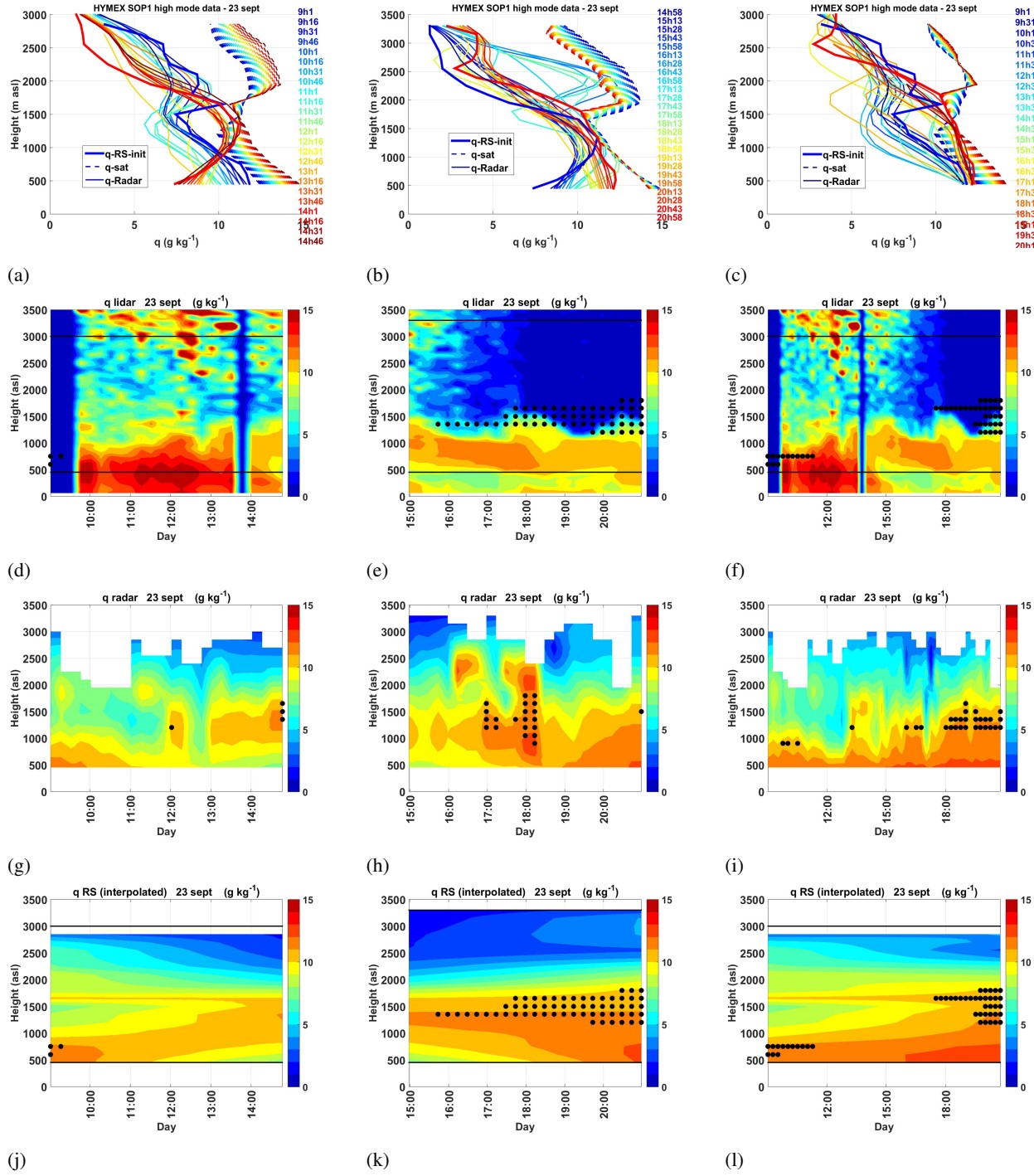

**Figure 8.** *Radar-based humidity profiles and humidity height-time cross-sections for the HYMEX 23 September case study: from RS1 (09:02 UTC, blue solid line in panel (a)) to RS2 (14:59 UTC, red solid line in panel (a)), from RS2 (blue, panel (b)) to RS3 (20:58 UTC, red solid line in panel (b)) and from RS1 (blue, panel (c)) to RS3 (red, panel (c)). First row is for the radar profiles (same details as in Fig. 6), second for the height-time humidity of the lidar, third for the radar and fourth for the interpolated RS. The dots in the last three rows demarcate the saturated values as in Fig. 7.*

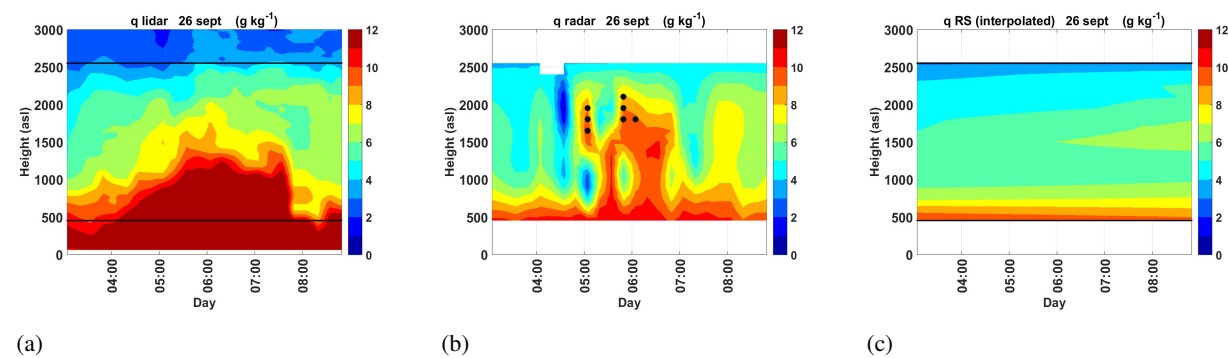

**Figure 9.** *Same as in Fig. 6 for 26 September 2012.*