# Peer review of "High resolution humidity profiles retrieved from wind profiler radar measurements"

_Atmospheric Measurement Techniques, 2017_

## Referee Comment (RC1) · Anonymous Referee #2 · 6 Oct 2017

**GENERAL COMMENTS**

This manuscript investigates the potential of UHF wind-profiling radar returns for the retrieval of atmospheric humidity profiles. Owing to the fundamental importance of water (in all phases) in the atmosphere, and to the fact that meteorological services are still primarily reliant on 12 hourly balloon-borne sensing, this technique would be very valuable if it could be used operationally. The technique itself is not new, and the development presented in the current manuscript is incremental rather than fundamental. Nevertheless, I think that the manuscript makes a useful contribution to this area of research. It is good that the authors have acknowledged instances where the technique does not produce reliable results.

SPECIFIC COMMENTS

The manuscript is clearly written and I have very few general or specific comments to make about it.

1) Section 2. The symbol q is usually used to imply the specific humidity (i.e. the mass of water vapour per unit mass of moist air) rather than the mixing ratio (r, the mass of water vapour per unit mass of dry air) as is stated at the beginning of this section. I suspect that q has simply been misnamed since the formulae in this section all appear to be consistent with it implying specific humidity. The abbreviation WVMR is used throughout the manuscript.

2) Although the figures are all sufficiently large to allow the important features to be seen, the labels are quite small on some of them - particularly Figs 9 - 12. It would be useful if the labels were made larger.

3) In connection with Figure 6, use of the gradient Richardson number will only identify regions of (dry) convective instability (which are expected to be confined to the boundary layer) and dynamic instability. I would expect moist convective processes to be a more significant contributor to turbulence at these altitudes, although such regions will not be identified in this way.

4) Page 17, line 27. "We checked the distributions of C\_n2 for the 3 datasets and found that the logarithmic averages of C\_n2 (close to the median values) gave 1.4, 31, and 1.0 10-14 m-2/3." Should the middle value be 31 or 3.1? The shown value of 31 is much larger then the two other values quoted.

5) Page 19, second paragraph. The authors discuss the possibility that the cloud connected with Fig. 9 may be virga. If so, the radar might be seeing hydrometeor scatter rather than clear-air scatter and the humidity retrieval algorithm would not be valid. This possibility should be discussed. This point is also relevant for the discussion of Fig. 10.

**TECHNICAL CORRECTIONS**
6) Page 18, line 18. The word "criterion" is misspelled as "criterium" twice in this section.

---

## Referee Comment (RC2) · Anonymous Referee #1 · 13 Nov 2017

The authors present and discuss a technique for retrieving vertical humidity profiles from both UHF radar measurements and radiosonde measurements. The general approach is well known and has been documented in several journal papers published during the last 30 years or so. The retrieval technique relies on quite a number of non-rigorous physical and meteorological assumptions and simplifications and is known to be not as robust as, e.g., the retrieval of wind velocities and of the refractive-index structure parameter. Improvements in our ability to retrieve humidity profiles from clear-air radar observations have been incremental, and a fundamental breakthrough is not to be expected.

The manuscript is much too long in comparison to its scientific content. If the paper contains results that might be worthy of publication, they are well hidden under a large

amount of unnecessary material. The figure captions are not self-explanatory. The abstract does not state the underlying physical hypotheses, assumptions, simplifications, and approximations, and it contains no information about the precision or accuracy of the retrieved humidity profiles. The conclusions section is much too long and does not present hard conclusions in a compelling and concise manner.

I recommend to reject the paper.

---

## Author Comment (AC2) · 11 Dec 2017

Color code :

*Comments from Referee 1 (RC2) (italic and blue)*

*Comments from Referee 2 (RC1) (italic and red)*

Changes we propose (magenta)

**Referee 1 (RC2) comments:**

*"The authors present and discuss a technique for retrieving vertical humidity profiles from both UHF radar measurements and radiosonde measurements. The general approach is well known and has been documented in several journal papers published during the last 30 years or so. The retrieval technique relies on quite a number of nonrigorous physical and meteorological assumptions and simplifications and is known to be not as robust as, e.g., the retrieval of wind velocities and of the refractive-index structure parameter. Improvements in our ability to retrieve humidity profiles from clear-air radar observations have been incremental, and a fundamental breakthrough is not to be expected.*

*The manuscript is much too long in comparison to its scientific content. If the paper contains results that might be worthy of publication, they are well hidden under a large amount of unnecessary material. The figure captions are not self-explanatory. The abstract does not state the underlying physical hypotheses, assumptions, simplifications, and approximations, and it contains no information about the precision or accuracy of the retrieved humidity profiles. The conclusions section is much too long and does not present hard conclusions in a compelling and concise manner.*

*I recommend to reject the paper."*

**Reply to Referee 1 (RC2)**

Despite the Referee's final recommendation to reject the paper , we have decided to move forward and proceed with the submission of a revised version of the manuscript. There are two main motivations behind this decision.

First of all, we have contacted the Associate Editor for an advice, and the Editor has encouraged us to submit a revised version of our manuscript. The Editor believes that the comments from both Referees can help us to improve the manuscript and has invited us to take them all under serious consideration and move forward in the process, which is what we are presently doing.

Second motivation: we have the impression that the final evaluation from the referee (rejection) is somewhat conflicting with the reminder part of the reviewer evaluation. In fact, besides some preliminary statement on the originality of the paper ("The general approach is well known and has been documented in several journal papers published during the last 30 years or so.") and the considered assumptions ("The retrieval technique relies on quite a number of non-rigorous physical and meteorological assumptions and simplifications"), the reviewer is stating that: ... "The manuscript is much too long in comparison to its scientific content. If the paper contains results that might be worthy of publication, they are well hidden under a large amount of unnecessary material." T**his is indeed an aspect that we can easily address by shortening and partially reshuffling the paper, that is what we are decided to do and are presently doing.** The reviewer is also stating that: ..."The figure captions are not self-explanatory." Again, **this is an aspect we can easily address and we actually already did it in the revised version of the paper**. The reviewer adds: ... "The abstract does not state the underlying physical hypotheses, assumptions, simplifications, and approximations, and it contains no information about the precision or accuracy of the retrieved humidity profiles. Again, **we added all** this **missing information in the revised version of the Abstract**. And again: ... "The conclusions section is much too long and does not present hard conclusions in a compelling and concise manner." **In the revised version of the paper, the conclusions have been shortened and the**

**text has been partially rewritten in order to present the conclusions in a compelling and concise manner.**

To summarize, **we believe that all issues raised by the Referee can be addressed (and have actually already been addressed in the revised version of the paper that we are in the process to submit)** and, consequently, the statement "I recommend to reject the paper" is somewhat conflicting with the specific points raised by the Referee, which can indeed be addressed.

With this premise, we would like to address (below) the different issues raised by the Referee, hoping that our replies can make him/her re-consider his/her judgment on the manuscript publication.

1 -  *"The general approach is well known and has been documented in several journal papers published during the last 30 years or so.* […] *Improvements in our ability to retrieve humidity profiles from clear-air radar observations have been incremental, and a fundamental breakthrough is not to be expected."*

We do not claim that this research is new and we agree with the comment from Referee 2, who underlined that *"The technique itself is not new, and the development presented in the current manuscript is incremental rather than fundamental."* We insisted on this point in the manuscript by quoting the authors who proposed or used the method and we made this clear in page 4, lines 12-13: 'Following Tsuda et al. (2001), Furumoto et al. (2006), Klaus et al. (2006) and Imura et al. (2007) also used Eq. (12) to compute humidity profiles. We use this equation in the present work.'

Nevertheless we point out that most research efforts that were carried out on this subject indicated that the method was promising for an operational use, although, to our knowledge, no work had been presented yet for such an application (page 1, line 23: 'However, as far as we know, no successful attempt to apply this method to operational observations has been reported in literature.'). That is why our aim was to check whether an operational application could be implemented or not. We tried to do so, improved the algorithm (by introducing the use of a transition level and a calibration process that varied with time and height) and obtained mildly encouraging results. Our objective was to honestly present the encountered difficulties and provide conclusive statements on the applicability of  the method.  This seems to have been appreciated by Referee 2, who wrote: *"It is good that the authors have acknowledged instances where the technique does not produce reliable results."*

2 -  '*The retrieval technique relies on quite a number of nonrigorous physical and meteorological assumptions and simplifications and is known to be not as robust as, e.g., the retrieval of wind velocities and of the refractive-index structure parameter.' […]*

We agree with you on the lack of robustness of the considered retrieval technique, and we propose to highlight the difficulties in the abstract by writing: **"The retrieval of humidity profiles from wind profiler radars has already been documented in the past 30 years and is known to be neither straightforward nor as robust as the retrieval of wind velocity, which exploits a physical property of electromagnetic waves (i.e. the Doppler effect). The main constraint to retrieve the humidity profile is the necessity to combine measurements from the wind profiler and additional measurements (such as observations from radiosoundings at a coarser time resolution). Furthermore, the method relies on some assumptions and simplifications that restrict the scope of its application. The first objective of this paper is to identify the obstacles and limitations and try to solve them, or at least define the field of applicability of the method. To improve the method, we propose to use the radar capacity to detect transition levels, such as the top level of the boundary layer, marked by a maximum in the radar reflectivity and to use this level as a new constraint for the algorithm** which  reduced the error affecting the specific humidity profile retrieval, with the mean bias never exceeding 0.25 g kg-1. **The second objective is to explore the capability of the**

**algorithm to retrieve the humidity vertical profiles for an operational purpose by comparing the results with observations from a Raman lidar."**

3 -   *The abstract does not state the underlying physical hypotheses, assumptions, simplifications, and approximations, and it contains no information about the precision or accuracy of the retrieved humidity profiles.*

The Abstract has been modified in the direction to include the limitations of the approach used in the paper. The following new sentences have been introduced: "The retrieval of humidity profiles from wind profiler radars has already been documented in the past 30 years and is known to be neither straightforward nor as robust as the retrieval the wind velocity. The main constraint to retrieve the humidity profile is the necessity to combine measurements from the wind profiler and additional measurements (such as observations from radiosoundings at a coarser time resolution). Furthermore the method relies on some assumptions and simplifications that restrict the scope of its application. **The first objective of this paper is to identify the obstacles and limitations and try to solve them, or at least define the field of applicability of the method.**" An estimate of the accuracy of the retrieved humidity profiles has also been introduced, the corresponding sentence of the Abstract now reading: " … **and to use this level as a new constraint for the algorithm** which  reduced the error affecting the specific humidity profile retrieval, with the mean bias never exceeding 0.25 g kg-1. "

*The conclusions section is much too long and does not present hard conclusions in a compelling and concise manner.*

[revised manuscript text omitted]

4 - *The manuscript is much too long in comparison to its scientific content. If the paper contains results that might be worthy of publication, they are well hidden under a large amount of unnecessary material.*

We could shorten the manuscript by removing several parts and shortening others. Specifically, we could remove the:
- description of case studies and the associated figure (4) : section 4.2, page 13, lines 10 -34 + page 14, lines 1-22
- discussion of the BLLAST upper layer:  page 15, lines 3-10
- discussion on the use of a constant calibration coefficient: p 17, lines 13-17
- last case study in section 5.2 : p 21, lines 19-34 and p 22, lines 1-12 and Fig. 12.
- Fig. 6 and 7 could be presented in one single Fig (with two parts). Total number of Fig. would be 9.
Additionally, we reshuffled the conclusion as proposed in point 3.

5 - *The figure captions are not self-explanatory.*

As suggested by the reviewer, we tried to improve figure captions in order to make them self-explanatory.

The caption of figure 1 now reads: "Vertical profiles of refractivity gradient (panels (a) and (b)) and humidity (panels (c) and (d)) using a systematic negative sign for M (panels (a) and (c)), or after assigning to M the sign of M provided by the RS observations (panels (b) and (d)). RS values (q-RS) are red solid  lines and radar values (q-Radar) are black solid lines. The  thin red lines (int-q-RS) identify the humidity profiles retrieved from the integration of $M_{RS}$ after averaging the RS observations by slices of 75 m, to match the vertical resolution of the radar. The red dashed  line (qsat-RS) is for the saturated humidity  profile. The horizontal dashed line delineates Hlim, the transition level used to separate the upper and lower part of the profile. The two values of $\alpha^2$ (calibration coefficients), over or below Hlim are also indicated."

The caption of figure 2 now reads: "Radar turbulence structure parameter $Cn^2$ (panels (a) and (d)) and humidity profiles (same details as in Fig. 1 (c)) for different Hlim levels. Panels (b) and (e) illustrate the WPR specific humidity profile (q-Radar) obtained considering the value of Hlim corresponding to the dominant $Cn^2$ peak observed in panels (a) and (d), respectively. Panels (c) and (f) illustrate the WPR specific humidity profile obtained considering the value of Hlim corresponding to a different relative maximum of the $Cn^2$ profile. In panels (b), (c), (e) and (f) the RS specific humidity profile (q-RS), the saturation specific humidity profile (qsat-RS) obtained from RS pressure and temperature profiles, and the saturation specific humidity profile (int-q-RS) obtained by integrating RS vertical gradient of refractivity ($M_{RS}$) are also indicated. Panels (a), (b) and (c) are for the high mode (vertical resolution 375 m, interpolated every 150 m), while panels (d), (e) and (f)  are for the low mode (vertical resolution 150 m)."

The caption of figure 3 now reads: "Specific humidity profiles obtained: i) using an automatic detection of the initial specific humidity value ($q_o$) at the level $z_o$ where the integration is initialized (panels (a) and (c)); ii)  through an adjustment of $q_o$ (panels (b) and (d)) for two different profiles from the BLLAST data set. The vertical resolution is 75 m. Same details as in Fig. 1 (c).

The caption of figure 4 now reads: "Panels (a), (c), (e) and (g) : scatterplots of radar versus RS specific humidity during June and July 2011 (BLLAST), September 2012n(HyMeX SOP1), October 2012 (HyMeX SOP1) and February 2013 (HyMeX SOP2), respectively, with the linear regression line (in red) and the 1:1 slope line (in black). The $R^2$ correlation coefficient of the regression is also specified. Panels (b), (d), (f) and (h) : Deviation profiles between the RS specific humidity measurements and radar-based estimate for the above specified data sets ± the standard deviation. The mean bias and standard deviation values for the whole dataset are reported, along with the maximum standard deviation value (g $kg^{-1}$).

---

## Author Comment (AC3) · 12 Dec 2017

The comment was uploaded in the form of a supplement:
https://www.atmos-meas-tech-discuss.net/amt-2017-265/amt-2017-265-AC3-supplement.pdf

---

## Author Response (AR1)

Color code :

*Comments from Referee 1 (RC2) (italic and blue)*

*Comments from Referee 2 (RC1) (italic and red)*

Changes we propose (magenta)

V1 is the previous manuscript version, V2 is the new manuscript one.

**Referee 1 (RC2) comments:**

*"The authors present and discuss a technique for retrieving vertical humidity profiles from both UHF radar measurements and radiosonde measurements. The general approach is well known and has been documented in several journal papers published during the last 30 years or so. The retrieval technique relies on quite a number of nonrigorous physical and meteorological assumptions and simplifications and is known to be not as robust as, e.g., the retrieval of wind velocities and of the refractive-index structure parameter. Improvements in our ability to retrieve humidity profiles from clear-air radar observations have been incremental, and a fundamental breakthrough is not to be expected.*

*The manuscript is much too long in comparison to its scientific content. If the paper contains results that might be worthy of publication, they are well hidden under a large amount of unnecessary material. The figure captions are not self-explanatory. The abstract does not state the underlying physical hypotheses, assumptions, simplifications, and approximations, and it contains no information about the precision or accuracy of the retrieved humidity profiles. The conclusions section is much too long and does not present hard conclusions in a compelling and concise manner.*

*I recommend to reject the paper."*

**Reply to Referee 1 (RC2)**

Despite the Referee's final recommendation to reject the paper , we have decided to move forward and proceed with the submission of a revised version of the manuscript. There are two main motivations behind this decision.

First of all, we have contacted the Associate Editor for an advice, and the Editor has encouraged us to submit a revised version of our manuscript. The Editor believes that the comments from both Referees can help us to improve the manuscript and has invited us to take them all under serious consideration and move forward in the process, which is what we are presently doing.

Second motivation: we have the impression that the final evaluation from the referee (rejection) is somewhat conflicting with the reminder part of the reviewer evaluation. In fact, besides some preliminary statement on the originality of the paper ("The general approach is well known and has been documented in several journal papers published during the last 30 years or so.") and the considered assumptions ("The retrieval technique relies on quite a number of non-rigorous physical and meteorological assumptions and simplifications"), the reviewer is stating that: ... "The manuscript is much too long in comparison to its scientific content. If the paper contains results that might be worthy of publication, they are well hidden under a large amount of unnecessary material." T**his is indeed an aspect we addressed by shortening and partially reshuffling the paper.** The reviewer is also stating that: ..."The figure captions are not self-explanatory." Again, **this is an aspect we addressed in the revised version of the paper**. The reviewer adds: ... "The abstract does not state the underlying physical hypotheses, assumptions, simplifications, and approximations, and it contains no information about the precision or accuracy of the retrieved humidity profiles. Again, **we added all** this **missing information in the revised version of the Abstract**. And again: ... "The conclusions section is much too long and does not present hard conclusions in a compelling and concise manner." **In the revised**

**version of the paper, the conclusions have been shortened and the text has been partially rewritten in order to present the conclusions in a compelling and concise manner.**

To summarize, **we think we have been able to address all issues raised by the Referee.**

With this premise, we would like to address (below) the different issues raised by the Referee, hoping that our replies can make him/her re-consider his/her judgment on the manuscript publication.

1 - *"The general approach is well known and has been documented in several journal papers published during the last 30 years or so. […] Improvements in our ability to retrieve humidity profiles from clear-air radar observations have been incremental, and a fundamental breakthrough is not to be expected."*

We do not claim that this research is new and we agree with the comment from Referee 2, who underlined that *"The technique itself is not new, and the development presented in the current manuscript is incremental rather than fundamental."* We insisted on this point in the manuscript by quoting the authors who proposed or used the method and we made this clear in page 4, lines 12-13 in V1, lines 16-17 in V2 : 'Following Tsuda et al. (2001) also, Furumoto et al. (2006), Klaus et al. (2006) and Imura et al. (2007) used Eq. (12) to compute humidity profiles. We use this equation in the present work.'

Nevertheless we point out that most research efforts that were carried out on this subject indicated that the method was promising for an operational use, although, to our knowledge, no work had been presented yet for such an application (page 1, lines 23-24 in V1; page 2, lines 2-3 in V2 : 'However, as far as we know, no successful attempt to apply this method to operational observations has been reported in literature.'). That is why our aim was to check whether an operational application could be implemented or not. We tried to do so, improved the algorithm (by introducing the use of a transition level and a calibration process that varied with time and height) and obtained mildly encouraging results. Our objective was to honestly present the encountered difficulties and provide conclusive statements on the applicability of the method. This seems to have been appreciated by Referee 2, who wrote: "It is good that the authors have acknowledged instances where the technique does not produce reliable results."

2 - *'The retrieval technique relies on quite a number of nonrigorous physical and meteorological assumptions and simplifications and is known to be not as robust as, e.g., the retrieval of wind velocities and of the refractive-index structure parameter.' […]*

We agree with you on the lack of robustness of the considered retrieval technique, and we propose to highlight the difficulties in the abstract by writing: "The retrieval of humidity profiles from wind profiler radars has already been documented in the past 30 years and is known neither to be straightforward and nor as robust as the retrieval of the wind velocity. The main constraint to retrieve the humidity profile is the necessity to combine measurements from the wind profiler and additional measurements (such as observations from radiosoundings at a coarser time resolution). Furthermore the method relies on some assumptions and simplifications that restrict the scope of its application. The first objective of this paper is to identify the obstacles and limitations and solve them, or at least define the field of applicability. To improve the method, we propose to use the radar capacity to detect transition levels, such as the top level of the boundary layer, marked by a maximum in the radar reflectivity. This forces the humidity profile from the free troposphere and from the boundary layer to coincide at this level, after an optimization of the calibration coefficients, and reduces the error. The resulting mean bias affecting the specific humidity profile never exceeds 0.25 g kg-1. The second objective is to explore the capability of the algorithm to retrieve the humidity vertical profiles for an operational purpose by comparing the results with observations from a Raman lidar."

3 -    *The abstract does not state the underlying physical hypotheses, assumptions, simplifications, and approximations, and it contains no information about the precision or accuracy of the retrieved humidity profiles.*

The Abstract has been modified in the direction to include the limitations of the approach used in the paper. The following new sentences have been introduced: "The retrieval of humidity profiles from wind profiler radars has already been documented in the past 30 years and is known to be neither straightforward nor as robust as the retrieval the wind velocity. The main constraint to retrieve the humidity profile is the necessity to combine measurements from the wind profiler and additional measurements (such as observations from radiosoundings at a coarser time resolution). Furthermore the method relies on some assumptions and simplifications that restrict the scope of its application. **The first objective of this paper is to identify the obstacles and limitations and try to solve them, or at least define the field of applicability of the method.**" An estimate of the accuracy of the retrieved humidity profiles has also been introduced, the corresponding sentence of the Abstract now reading: The resulting mean bias affecting the specific humidity profile never exceeds 0.25 g kg-1 "

*The conclusions section is much too long and does not present hard conclusions in a compelling and concise manner.*

[revised manuscript text omitted]

4 - *The manuscript is much too long in comparison to its scientific content. If the paper contains results that might be worthy of publication, they are well hidden under a large amount of unnecessary material.*

We shortened the manuscript by removing several parts and shortening others. Specifically, we removed the:
- description of case studies and the associated figure (4) : section 4.2, page 13, lines 10 -34 + page 14, lines 1-22 from V1.
- discussion of the BLLAST upper layer:  page 15, lines 3-10 from V1.
- discussion on the use of a constant calibration coefficient: p 17, lines 13-17 from V1.
- last case study in section 5.2 : p 21, lines 19-34 and p 22, lines 1-12 and Fig. 12 from V1.
- Fig. 6 and 7 could be presented in one single Fig (with two parts). Total number of Fig. is now 9.

Additionally, we reshuffled the conclusion as proposed in point 3.

5 - *The figure captions are not self-explanatory.*
As suggested by the reviewer, we tried to improve figure captions in order to make them self-explanatory.

The caption of figure 1 now reads: "Vertical profiles of refractivity gradient (panels (a) and (b)) and humidity (panels (c) and (d)) using a systematic negative sign for M (panels (a) and (c)), or after assigning to M the sign of M provided by the RS observations (panels (b) and (d)). RS values (q-RS) are red solid lines and radar values (q-Radar) are black solid lines. The thin red lines (int-q-RS) identify the humidity profiles retrieved from the integration of $M_{RS}$ after averaging the RS observations by slices of 75 m, to match the vertical resolution of the radar. The red dashed line (qsat-RS) is for the saturated humidity profile. The horizontal dashed line delineates Hlim, the transition level used to separate the upper and lower part of the profile. The two values of $\alpha^2$ (calibration coefficients), over or below Hlim are also indicated."

The caption of figure 2 now reads: "Radar turbulence structure parameter $Cn^2$ (panels (a) and (d)) and humidity profiles (same details as in Fig. 1 (c)) for different Hlim levels. Panels (b) and (e) illustrate the WPR specific humidity profile (q-Radar) obtained considering the value of Hlim corresponding to the dominant $Cn^2$ peak observed in panels (a) and (d), respectively. Panels (c) and (f) illustrate the WPR specific humidity profile obtained considering the value of Hlim corresponding to a different relative maximum of the $Cn^2$ profile. In panels (b), (c), (e) and (f) the RS specific humidity profile (q-RS), the saturation specific humidity profile (qsat-RS) obtained from RS pressure and temperature profiles, and the saturation specific humidity profile (int-q-RS) obtained by integrating RS vertical gradient of refractivity ($M_{RS}$) are also indicated. Panels (a), (b) and (c) are for the high mode (vertical resolution 375 m, interpolated every 150 m), while panels (d), (e) and (f) are for the low mode (vertical resolution 150 m)."

The caption of figure 3 now reads: "Specific humidity profiles obtained: using an automatic detection of the initial specific humidity value ($q_o$) at the level $z_o$ where the integration is initialized (panels (a) and (c)); through an adjustment of $q_o$ (panels (b) and (d)) for two different profiles from the BLLAST data set. The vertical resolution is 75 m. Same details as in Fig. 1 (c).

The caption of figure 4 now reads: "Panels (a), (c), (e) and (g) : scatterplots of radar versus RS specific humidity during June and July 2011 (BLLAST), September 2012 (HyMeX SOP1), October 2012 (HyMeX SOP1) and February 2013 (HyMeX SOP2), respectively, with the linear regression line (in red) and the 1:1 slope line (in black). The $R^2$ correlation coefficient of the regression is also specified. Panels (b), (d), (f) and (h) : deviation profiles between the RS specific humidity measurements and radar-based estimate for the above specified data sets ± the standard deviation. The mean bias and standard deviation values for the whole dataset are reported, along with the maximum standard deviation value (g kg$^{-1}$).

**Reply to Referee 2 (RC1)**

We thank you for the detailed review you provided. We tried to take into account your suggestions and think they will help to clarify and improve the paper. Your comments are in italic. We highlighted in red the modifications we propose to do in the manuscript. V1 refers to the previous manuscript and V2 to the new version of the manuscript.

*'1) Section 2. The symbol q is usually used to imply the specific humidity (i.e. the mass of water vapour per unit mass of moist air) rather than the mixing ratio (r, the mass of water vapour per unit mass of dry air) as is stated at the beginning of this section. I suspect that q has simply been misnamed since the formulae in this section all appear to be consistent with it implying specific humidity. The abbreviation WVMR is used throughout the manuscript.'*

Although r and q are very similar, we changed WVMR for q, specific humidity, throughout the figures and whole manuscript. Concurrently we propose to change the q definition (top of page 3, at the beginning of the 'Theoretical background' section) as: ' We can also express N in terms of specific humidity, q (kg of water vapor per kg of moist air). q is the parameter we aim at retrieving in the present work. Using the approximation q=0.622 e/ (P-0.378e) ≈ 0.622 e/P, Eq. (1) becomes:…', and to insert : 'In the following, we will use symbol q for specific humidity and water vapor mixing ratio equally, since the percentage deviation between both is rarely exceeding 1%, even in case of large humidity concentrations, which remains far less than the systematic and statistical uncertainties affecting the lidar mixing ratio measurements.' in section 3.2 (p9, line 6 in V1, or line 15 in V2), where we describe the lidar measurements. Anyway, the approximation that is done above (where 0.378 e is considered as small relative to P), implies that the accuracy of the humidity retrieval by the radar cannot be better than the difference between q and r, since 0.622 e/P is also an approximation of r.

*'2) Although the figures are all sufficiently large to allow the important features to be seen, the labels are quite small on some of them - particularly Figs 9 - 12. It would be useful if the labels were made larger.'*

New figures are provided with larger labels.

*'3) In connection with Figure 6, use of the gradient Richardson number will only identify regions of (dry) convective instability (which are expected to be confined to the boundary layer) and dynamic instability. I would expect moist convective processes to be a more significant contributor to turbulence at these altitudes, although such regions will not be identified in this way.'*

We agree that the link between the gradient Richardson number and the calibration coefficients is relevant for the dry conditions of BLLAST, but not for those of HYMEX SOP1, for which the boundary layer was not very deep (most of the time), nor for those of HYMEX SOP2, for which dynamic turbulence systematically prevailed over thermal turbulence.

To be clearer in the manuscript, after the sentence: ' The same analysis was applied to the HyMeX datasets from which no significant result arose, neither during SOP1 nor SOP2 (not shown)' (p16, line 29 in V1), we removed the sentence: 'Unstable conditions were too occasional to show any tendency. Usually, the upper layer exhibited a stronger stratification than the lower layer, but the calibration coefficients varied irrespectively of this variation.' and replaced it by: 'During HyMeX, the development of the boundary layer was most of the time generated by mechanical turbulence (due to the wind intensity or to the roughness change at the sea/land transition). This can explain why the gradient Richardson number is not a good indicator of the variability in the calibration coefficients under the HyMeX conditions. There is also no clear difference in the HyMeX coefficients between the lower and upper parts of the profiles, probably because moist convection equally affected all levels.' (p15, line 10 in V2).
The fact is that the calibration coefficients for HYMEX during both SOPs do not vary as much as those during BLLAST. It is a hypothesis to consider that the difference in the variability could result from the

buoyancy contribution due to the temperature fluctuations. The point is that turbulence is also taken into account in $C_n^2$ through $\varepsilon$ (Eq. 15), which includes contributions from temperature, wind and moist (through the release of latent heat). We recognize we did not address this issue and kept a practical point of view by underlining that the calibration coefficients may vary, and recommending the use of a calibration coefficient that varies with time and space.

This conclusion appeared in V1 in section 4.3, p17, line 8 ('The main conclusion we can draw from these results,  coming primarily from the BLLAST data, is the necessity of  distinguishing between the mixed layer and the free troposphere in case of unstable conditions in the low troposphere.') and we added in the final section of V2 (page 20, line 23) : 'We highlighted the necessity of calibrating the vertical gradient of refractivity provided by the radar, with calibration coefficients likely to vary in time and space.'

*'4) Page 17, line 27. "We checked the distributions of C_nˆ2 for the 3 datasets and found that the logarithmic averages of C_nˆ2 (close to the median values) gave 1.4, 31, and 1.0 10ˆ-14 mˆ-2/3." Should the middle value be 31 or 3.1? The shown value of 31 is much larger then the two other values quoted.'*

We maintain 31. This high value is the reason why we explained (p17, lines 29-32 in V1 and p16, lines 1-3 in V2) that HYMEX SOP1 conditions are borderline to apply the method (Bragg and not Rayleigh conditions). We are aware that the application of the method under these conditions is questionable, but these conditions are characteristic of the whole Mediterranean western basin, during at least 2 months every year, and probably characteristic of other regions in the world. It was worthy to assess the method under these specific conditions, especially because models had difficulties to retrieve accurate values of humidity under these conditions, which was one of the motivation to organize the experiment.

*'5) Page 19, second paragraph. The authors discuss the possibility that the cloud connected with Fig. 9 may be virga. If so, the radar might be seeing hydrometeor scatter rather than clear-air scatter and the humidity retrieval algorithm would not be valid. This possibility should be discussed. This point is also relevant for the discussion of Fig. 10.'*

We agree with you: they cannot be virga, since the turbulence structure parameters, as well as the vertical velocity measured by the radar,  would be higher if it was a virga, and we could not have applied the method. We intend to replace 'virga'  by  'cloud' in the new version of the manuscript.

*'TECHNICAL CORRECTIONS*
*C2*
*6) Page 18, line 18. The word "criterion" is misspelled as "criterium" twice in this section.'*

It has been corrected in V2.

[revised manuscript text omitted]

The previous illustration revealed one major limitation of the approach used to retrieve the humidity profile. In fact, in case of multiple strong inversions in the low troposphere, as those observed in Fig. 3, the method failed since we had no possibility to distinguish between the different sublayers. In this specific case, the inversion located at 2700 m cannot be retrieved since it is situated at the upper boundary of the radar height range.

[revised manuscript text omitted]

---

## Author Response (AR2)

**Reply to Referee 2**

Thanks a lot for these additional suggestions to improve the paper. You will find below the modifications we did accordingly.

**1)	In relation to footnote 1 on page 2, if the authors have a suitable reference to justify that only raindrops larger than 10 um can be detected by UHF wind profiling radars, they should include it for completeness.**

10um was underestimated. The boundary between cloud air and rain should be closer to 100um than to 10um. The rainfall detection threshold is determined by the value of radar reflectivity factor where the Rayleigh scattering from precipitation becomes stronger than the Bragg scattering from clear air (Ralph, 1995). It depends on the radar wavelength and at constant wavelength, it depends on the particles size. Cloud particles have too small reflectivity factors to be detected by UHFs.
According to Ralph (1995), if we consider a terminal velocity of the rain of 0.5 m/s, the raindrops size would be around 100um. 0.5 m/s is also the threshold in the Doppler power spectrum that allows to distinguish between the peak of clear air and the peak of Rayleigh scattering (assuming that the vertical velocity of the air is negligible).
We do not wish to include these details in the text, but we followed your suggestion and quoted Ralph (1995). And we replaced 10 um by 100 um.

**2) In relation to equation 14 on page 5, I think that dV/dz should represent the vertical shear of the of the horizontal wind vector rather than of the horizontal wind speed. The authors have correctly used the former for equation 18, page 14.**

You are totally right and we made the change p 5 and consequently p 14.

**3) Use of the word "inversion" on line 8 of page 13 is confusing since it is applied to sharp negative gradients of specific humidity. Although the feature at 3000 m in Figure 3a does appear to correspond to a temperature inversion (the relevant part of the q-sat curve is obscured by the legend), the one at 4400 m in Figure 3c does not.**

The word 'inversion' was not only confusing but also wrongly used: our aim was not to comment the temperature inversion but instead the sharp decrease of the moisture slope (these decreases were not necessary linked to temperature inversions). So we removed the word 'inversion' and replaced it by something more appropriate (throughout the whole paper).

**4) In equations 1, 4, 5, 6, 8, and 12, as well as in the inline equation on line 15 of page 4, a multiplication sign should be used to separate the mantissa and exponent portions of the coefficients - e.g. in equation 1, 3.73 x 10^5 should be used instead of 3.73 10^5. I realise that there is a small gap between the two terms, but in most cases the gap is so small that the above reads as 3.7310^5.**

We enlarged the gaps since we do not find that the alternative (use of character '\times' in Latex) would be nice. We hope it is clearer now.

**5) line 15 of page 6. This is the first time that the abbreviation ABL has been used. It should be defined here rather than on page 7. Similarly, AGL should be defined on line 11 of page 7 and ASL on line 7 of page 11.**

Done, thanks.

**6) In the legend for Figure 5, it is difficult to distinguish between the shapes used for "lower layer" and "upper layer" owing to the pale green used to illustrated them.**

We added an edge to the symbols, including those of the legend. The shapes are also described in the caption.